# HexGen-2: Disaggregated Generative Inference of LLMs in Heterogeneous Environment

**Youhe Jiang,**\* **Ran Yan,**\* **Binhang Yuan**†
Department of Computer Science and Engineering
The Hong Kong University of Science and Technology

## Abstract

Disaggregating the prefill and decoding phases represents an effective new paradigm for generative inference of large language models (LLM), which eliminates prefill-decoding interference and optimizes resource allocation. However, it is still an open problem about how to deploy the disaggregated inference paradigm across a group of heterogeneous GPUs, which can be an economical alternative to deployment over homogeneous high-performance GPUs. Towards this end, we introduce HexGen-2, a distributed system for efficient and economical LLM serving on heterogeneous GPUs following the disaggregated paradigm. Built on top of HexGen, the core component of HexGen-2 is a *scheduling algorithm* that formalizes the allocation of disaggregated LLM inference computations and communications over heterogeneous GPUs and network connections as a constraint optimization problem. We leverage the *graph partitioning* and *max-flow* algorithms to co-optimize resource allocation, parallel strategies for distinct inference phases, and the efficiency of inter-phase key-value (KV) cache communications. We conduct extensive experiments to evaluate HexGen-2, i.e., on OPT (30B) and LLaMA-2 (70B) models in various real-world settings, the results reveal that HexGen-2 delivers up to a $2.0\times$ and on average a $1.3\times$ improvement in serving throughput, reduces the average inference latency by $1.5\times$ compared with state-of-the-art systems given the same price budget, and achieves comparable inference performance with a 30% lower price budget.

## 1 Introduction

Large Language Models (LLMs), such as OPT (Zhang et al., 2022), LLaMA (Touvron et al., 2023), GPT (OpenAI, 2024), Gemini (Reid et al., 2024), Claude (Anthropic, 2024) and Mixtral (Jiang et al., 2024a) have shown exceptional performance across various advanced applications. However, deploying the generative inference service for such LLMs can be costly, typically requiring a substantial number of homogeneous, high-performance GPUs to meet the service demands, such as first token latency and generation throughput. In this paper, we explore an alternative solution that *deploys the most advanced disaggregated generative inference paradigm over a set of heterogeneous GPUs to provide an efficient and economical LLM service*.

Disaggregated inference is currently the most *efficient* framework for serving the generative inference requests of LLMs (Zhong et al., 2024; Patel et al., 2024). By splitting the prefill phase (compute-bounded) and decoding phase (HBM IO-bounded) across different GPUs, the disaggregation significantly reduces interference between different requests and enables more flexible parallel configurations for the two phases. When compared with colocating the prefill and decoding computations, the disaggregated approach optimizes resource usage and enhances the scalability and efficiency of the LLM inference service. Recent efforts (Jiang et al., 2024b; Griggs et al., 2024; Zhao et al., 2024; Miao et al., 2024) have shown that serving LLMs with heterogeneous GPUs can be a *economical* alternative to deploying over homogeneous high-performance GPUs. Heterogeneous deployments offer significant opportunities to reduce inference service costs by leveraging the wide availability of diverse GPU types across commercial and private computing platforms. Note that Nvidia typically releases new GPU generations every 24 months, e.g., Turing in 2018, Ampere in

---

\*Equal contribution.
†Correspond to Binhang Yuan (`biyuan@ust.hk`).

2020, Hopper in 2022, and Blackwell scheduled for Q4 2024; but one particular version of GPU general remains in use for a much longer period.[1].

The wide availability of heterogeneous GPU pools presents significant opportunities to adapt the most advanced disaggregated inference paradigms. However, effectively adapting the disaggregated paradigm to this heterogeneous setting is much harder to implement than to ask for. Traditional implementation of co-locating prefill and decoding phases only leverage standard *tensor model parallelism* (Narayanan et al., 2021) and *pipeline parallelism* (Huang et al., 2019) for LLM inference, where only the activations are communicated. In the disaggregated paradigm, transferring the key-value (KV) cache between prefill and decoding model replicas introduces significant data movement, potentially creating a communication bottleneck that must be carefully managed in a heterogeneous setting. Additionally, the flexibility of parallel configurations among prefill and decoding model replicas also introduces new complexity in the heterogeneity-aware scheduling.

Towards efficiently adapting the disaggregated paradigm under the heterogeneous setting, we identify two types of new challenges and opportunities that previous heterogeneity-aware scheduling approaches (Jiang et al., 2024b) fail to integrate:

- **Accommodate the computation flexibility in disaggregated paradigm.** In a heterogeneous setting, each GPU type has distinct peak FLOPS, HBM memory bandwidth, and HBM memory limit, even making optimal computation allocation for the colocating paradigm a difficult problem. The disaggregated paradigm adds further complexity, as the prefill and decoding phases have different resource requirements and favor specific parallel strategies depending on varying LLM inference workloads, such as arrival rates and input/output sequence lengths.

- **Accommodate additional KV cache movement over heterogeneous connections.** GPU communication bandwidth also varies widely, from different NVLink and PCIe generations within a server to InfiniteBand(IB), RoCE, TCP, and Ethernet connections among different servers. Along with communication demands from parallel strategies within each model replica, disaggregated inference requires extensive KV cache transmissions, which are especially sensitive to low-bandwidth links. Therefore, an effective scheduling algorithm is essential to manage communication across heterogeneous GPU connections and minimize costs.

In order to overcome these challenges, we propose HEXGEN-2, a disaggregated LLM inference system that coordinates distributed LLM inference computations and communications over a set of GPUs with different computation capabilities and heterogeneous network connections. Our contributions are summarized as:

**Contribution 1:** We formulate the scheduling problem of allocating disaggregated LLM inference computations over a set of heterogeneous GPU devices as a constraint optimization problem. To solve this problem efficiently, we propose a sophisticated scheduling algorithm that employs a combination of graph partitioning and max-flow algorithm to coordinate the resource allocations and parallelism plans for the prefill and decoding phases of LLM inference. Concretely, the graph partitioning algorithm partitions the available GPUs into multiple model serving groups, where each group should be dedicated to serving a prefill or decoding model replica; and the max-flow algorithm guides the iterative refinement of the graph to optimize model placement.

**Contribution 2:** We implement HEXGEN-2, a heterogeneous LLM inference system that facilitates tensor model parallelism and pipeline parallelism with a disaggregated paradigm. HEXGEN-2 allows the two phases of LLM inference to be split onto separate GPUs with different parallel plans, effectively eliminating prefill-decoding interference and boosting inference performance.

**Contribution 3:** We evaluate HEXGEN-2 through extensive experiments, where we compare HEXGEN-2's system efficiency across various LLM inference workloads with HEXGEN on several heterogeneous settings and DISTSERVE on a standard homogeneous setting. We conduct these comparisons on the popular LLM models OPT (30B) and LLAMA-2 (70B). We show that given the same budget in terms of cloud service fees, HEXGEN-2 can choose to achieve up to a $2.0\times$ and on average a $1.3\times$ higher serving throughput or reduce the average inference latency by $1.5\times$. Additionally, when given only $70\%$ of the budget, HEXGEN-2 can still maintain a comparable level of inference service compared to the homogeneous baseline.

---

[1]For example, Tesla K80 GPUs, released in 2006, are still available on AWS as `p2` instances

## 2 PRELIMINARY

**LLM generative inference.** Given the input request, the LLM inference process typically contains two phases: *prefill* and *decoding*. The prefill phase processes the request to compute the KV cache and generates the first token for the response in a single step. The decoding phase then takes the last input token and KV cache as inputs to generate subsequent tokens by one token at each step. The distinct characteristics of both phases lead to differing GPU resource utilization: the prefill phase is compute-bound, whereas the decoding phase is HBM memory I/O-bound. Naive implementation of the inference engines colocates the two phases on the same group of GPUs, despite their distinct computational characteristics. Two standard strategies are applied to parallelize the LLM inference computation: *tensor model parallelism* and *pipeline parallelism*. Tensor model parallelism (TP) (Narayanan et al., 2021) distributes inference computations across multiple GPUs by partitioning the weight matrices of transformer layers both row-wisely and column-wisely, each layer's output activations are aggregated through two `AllReduce` operations. Pipeline parallelism (PP) (Huang et al., 2019) divides the model into multiple stages, each assigned to a specific GPU or group of GPUs for execution, the inter-layer activations are communicated between stages.

**Inference serving goal.** There are two essential metrics to evaluate LLM serving: *throughput* and *inference latency*. Throughput refers to the number of tokens a system can generate within a specified time period. Inference latency is the time required to complete each inference request from start to finish. We assess system performance on inference latency using service level objective (SLO) attainment, which gauges the proportion (e.g., $99\%$) of requests fulfilled within a time frame predefined by the SLO. This SLO is adjusted to various multiples of single device execution latency (termed as SLO scale) to measure performance under different degrees of SLO stringency.

**Batching.** Due to the computational difference of the prefill and decoding phases, integrating batching strategies leads to varying performance outcomes. As shown in Figure 1, in the prefill phase, a small batch size quickly saturates the GPU's computation capacity — Once the total number of batched tokens reaches $2048$, no further improvement in throughput is observed but the prefill latency escalates with batch size. Conversely, in the decoding phase, where the system bottleneck lies in scanning the LLM parameters, the throughput increases linearly as the total number of batched tokens rises, highlighting the effectiveness of batching in this phase for performance enhancement. The current state-of-the-art LLM serving system employs a batching optimization called *continuous batching* (Yu et al., 2022), which batches the prefill of new requests with the decoding of ongoing requests to enhance GPU utilization. However, this leads to severe prefill-decoding interference. Adding a single prefill job to a batch of decoding requests significantly slows down both processes, with the slowdown intensifying as the prefill length increases.

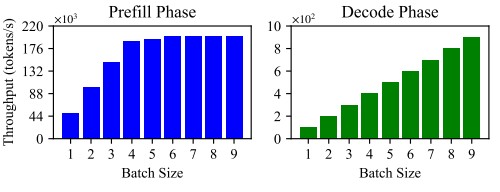

Figure 1: Effects of batching on different phases (LLAMA-2 (7B) inference with an input length of 512 on a single A100 GPU).

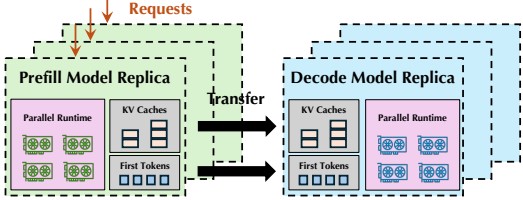

Figure 2: Illustration of disaggregated paradigm.

**Disaggregated architecture.** As the two phases in LLM inference have distinct characteristics, recent efforts (Zhong et al., 2024; Patel et al., 2024; Jin et al., 2024; Qin et al., 2024; Hu et al., 2024) propose a disaggregated inference architecture that splits the two phase in separate hardware resources. In the disaggregated inference architecture (See Figure 2), there are two types of model replicas: *prefill model replica* is responsible for taking the incoming request, generating the first token and KV cache; *decoding model replica* takes the generated token and KV cache as inputs, and generates the subsequent tokens. This separation enhances LLM serving by: (1) Eliminate the prefill-decoding interference; (2) Allow prefill and decoding model replicas to use different batching and parallelism strategies — Prefill replicas benefit from tensor model parallelism and smaller batches to reduce per-request latency, while decoding replicas perform better with larger batches to

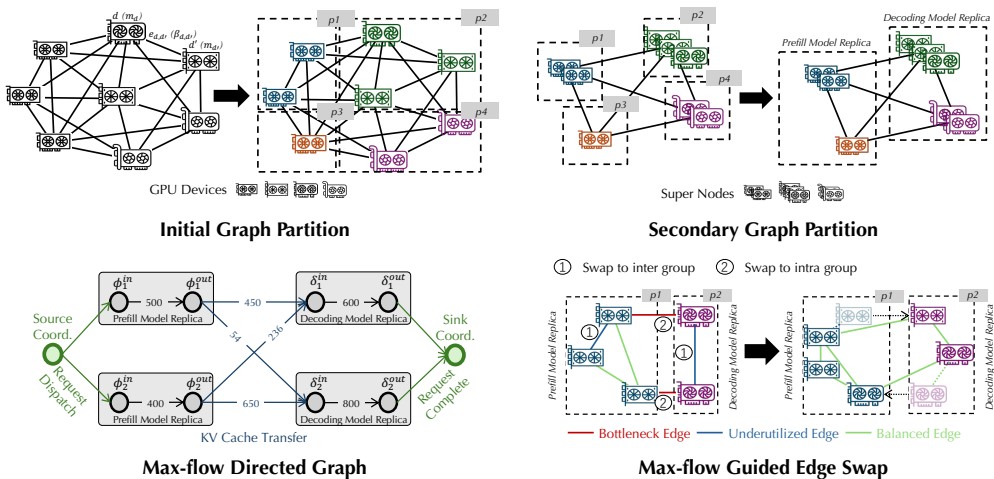

Figure 3: Illustration of each scheduling step.

maximize throughput. (3) Accommodate varying LLM serving workloads by adjusting resource allocations between the two phases, e.g., the coding workload characterized in (Patel et al., 2024) with typically longer prefill and shorter decoding sequence lengths requires more resources for prefill to optimize performance. As prefill and decode model replicas operate independently, it is crucial to transmit the KV cache from the prefill to the decode model replicas. Given the large volume of KV cache in LLM serving, current implementations necessitate a high-bandwidth communication link to facilitate the transmission of the KV cache. (Patel et al., 2024) utilize InfiniteBand (IB) for inter-node KV cache transmission, while (Qin et al., 2024) deploy their system on GPU clusters equipped with RDMA network cards, and (Zhong et al., 2024) collocate prefill and decode model replicas on GPUs within the same node to expedite KV cache transmission via NVLink. We also include the discussion of disaggregation versus chunked prefill in Appendix D.

## 3 SCHEDULING ALGORITHM IN HEXGEN-2

The core technique component in HEXGEN-2 is a scheduling module that can efficiently allocate the heterogeneous GPUs to serve prefill or decoding model replicas. In this section, we formulate the scheduling problem and introduce our solution.

### 3.1 PROBLEM FORMALIZATION

To support LLM serving with the disaggregated paradigm under heterogeneity, the scheduling algorithm should determine four essential allocations: (1) *the group partition*, i.e., how to partition the GPUs to multiple groups, where each responsible for serving one model replica; (2) *the group type*, i.e., whether a group serves a prefill or decoding model replica. (3) *the parallel strategy* for each model serving group, i.e., the combination of TP and PP under the heterogeneous setting (Jiang et al., 2024b); (4) *the KV cache communication strategy* among prefill and decoding model replicas. We term a solution to these four components as a *model placement strategy*.

Given the exponential search space, determining the optimal model placement is an NP-hard problem. To solve the problem, we adopt a two-phase search algorithm, which can be summarized as:

- **Graph partition**: Given a set of heterogeneous GPU devices **D**, the first phase (§3.2) aims to partition them into multiple model serving groups, and determines the group type.

- **Max flow**: Based on the outputs from the first phase, the second phase (§3.3) find the current optimal parallel strategies for prefill and decoding model replicas, and generates the optimal KV cache communication strategy among them.

- **Iterative refinement**: We iteratively repeat the two-phase algorithm to find the optimal model placement strategy (§3.4) that maximizes the end-to-end system performance.

## 3.2 First Phase: Graph Partition

The first phase of our scheduling algorithm aims to partition the GPU devices $\mathbf{D}$ into multiple model serving groups and determine whether each group is a prefill or decoding model replica. We first formulate the GPU device set $\mathbf{D}$ as a global graph $\mathrm{G} = (\mathbf{D}, \mathbf{E})$, with each GPU $d \in \mathbf{D}$ representing a *graph node*, and the GPU memory limit $m_d$ defined as the node weight. The communication link $e_{d,d'} \in \mathbf{E}$ between GPU $d$ and $d'$, $\forall d, d' \in \mathbf{D}$, is defined as the *graph edge*, with communication bandwidth $\beta_{d,d'}$ as the edge weight. Then, we partition the formulated graph $\mathrm{G}$ into partition $\mathbf{P} = \{p_1 \ldots p_K\}$, where $p_k$ denotes the $k$-th model serving group, and determine the type for each group. Concretely, there are three steps in the first phrase:

**Step (i) - Initial partition**: We first partition the global graph into multiple model serving groups based on edge weights (bandwidths), and balance the node weights (memory capacities) across groups. We leverage the *spectral partitioning* method (Alpert & Yao, 1995) to partition the graph $\mathrm{G}$ into $K$ groups, which uses the eigenvectors of the Laplacian matrix to guide partitioning and minimize inter-group edge weights. The group size $K$ is determined by dividing the cluster's total memory by the estimated memory required for a single model replica (detialed in Appendix A). Then we adopt the *Kernighan-Lin algorithm* (Kernighan & Lin, 1970) to iteratively refine the partition $\mathbf{P}$ by swapping node pairs between groups, which further reduces edge weights and balances node weights (memory capacities) across groups. Figure 3 demonstrates the process. Note that we balance memory rather than compute capacity to avoid potential OOM issues and provide a solid starting point for further optimization.

**Step (ii) - Coarsen & secondary partition**: We then determine the group type, where the graph is coarsened and partitioned again to determine the model replica type for each group. Note that coarsen is a common operation that merges nodes and edges to simplify graph partition (Hendrickson et al., 1995). Here, the coarsening operation merges graph nodes (GPUs) within the same group (model replica) into super nodes, which ensures the graph only includes relationships among the super nodes. The coarsened graph is then partitioned to distinguish between prefill and decoding model replicas. As illustrated in Figure 3, the four super nodes are divided into two parts: the two super nodes on the left are designated as prefill model replicas, while the two on the right are designated as decoding model replicas. Different from initial partition, the secondary partition focuses on *maximizing* inter-partition edge weights (i.e., the edge weights between prefill and decoding model replicas) to support frequent KV cache communications between different group types.

**Step (iii) - Projection**: Once we allocate the super nodes into prefill and decoding model replicas, we need to apply project operation, i.e., the reverse operation of the coarsen operation described in step (ii), to recover the GPU information within each super node. Note that after the projection, we can leave the problem of determining the optimal parallel strategies for each prefill or decoding model replica based on the GPU information within each super node during the second phase.

## 3.3 Second Phase: Max-flow

The second phase of our scheduling algorithm determines the parallel strategies within each super node and KV cache communication strategies between each super node. We leverage *max-flow*, as a promising method, to formulate the disaggregated inference pradiagm. Taking the partitioned graph from the first phase as input, we transform it into a *directed graph* with *compute nodes* and *network connections*. We define the source and sink of the directed graph to be the coordinator node $h$, which is responsible for request dispatching and completion. Formally, we define:

**Compute nodes.** The prefill and decoding model replicas are defined as compute nodes $\mathcal{C}$, with $\phi_i \in \mathcal{C}$ denoting a prefill model replica and $\delta_i \in \mathcal{C}$ denoting a decoding model replica. For each compute node $\phi_i/\delta_i \in \mathcal{C}$, we force it connect with two other nodes in the graph, named $\phi_i^{in}/\delta_i^{in}$ and $\phi_i^{out}/\delta_i^{out}$. The capacity of the directed edge $(\phi_i^{in}/\delta_i^{in}, \phi_i^{out}/\delta_i^{out})$ represents the maximum number of requests this node can process within a certain time period T (e.g., 10 minutes). We adopt the *inference cost model* from HEXGEN (Jiang et al., 2024b) and detail the node capacity estimation in Appendix A. To optimize capacity, the optimal parallel strategy should be selected for each node. As discussed in §2, given the distinct computational characteristics of different phases, their optimal parallel strategies also vary. For prefill model replicas, we aim to determine the *latency-optimal* parallel configurations, as they are computation-intensive and batching does not enhance efficiency. In contrast, for decoding model replicas, we aim to deduce the *throughput-optimal* parallel configurations, since this phase is memory I/O-bound and benefits from batching more requests. Based on these considerations, we iterate through all possible model parallelism

combinations for each model replica and select the optimal one. For compute node $\phi_i/\delta_i$, the amount of flow that passes through $(\phi_i^{in}/\delta_i^{in}, \phi_i^{out}/\delta_i^{out})$ should be no larger than its maximum capacity.

**Network connections.** A node in the directed graph might be connected with any other nodes, while only a subset of those connections are valid. A *valid connection* should satisfy one of the following criteria: ([1]) the connection is from coordinator node $h$ to compute node $\phi_i$, we represent the connection with directed edge $(source, \phi_i^{in})$; ([2]) the connection is from $\delta_i$ to coordinator node $h$, we represent the connection with directed edge $(\delta_i^{out}, sink)$; ([3]) the connection is from a compute node $\phi_i$ to another compute node $\delta_i$, we represent the connection with directed edge $(\phi_i^{out}, \delta_i^{in})$. The edge capacity equals the maximum number of requests this connection can process within the time period T. Note that for connection type ([3]), between any two prefill and decoding model replicas $\phi_i$ and $\delta_i$ with an edge connection, each GPU containing the $j$-th layer within $\phi_i$ should transmit its KV cache to the matching GPU housing the $j$-th layer within $\delta_i$. The edge capacity is determined by the collective performance of all GPU-to-GPU transmission connections, as each connection is responsible for a portion of the KV cache transmission. The estimation of edge capacity is detailed in Appendix A. We only permit flow to pass through valid network connections, and the transmitted flow should not exceed the maximum capacity of the connection.

After constructing the directed graph, we run *preflow-push algorithm* (Cheriyan & Maheshwari, 1989) to get the max flow between source and sink node, with one unit of flow representing one request that can pass through a compute node or network connection per unit time. This algorithm continuously pushes the maximum allowable flow up to the edge's capacity to maximize the flow through the direct connection. The generated *flow assignments* between compute nodes $\phi_i$ and $\delta_i$ are used to guide the KV cache communication. The communication frequency is set to be proportional to these flow values to follow the max flow of the directed graph without creating bursts, as illustrated in Figure 3. However, the algorithm may not fully utilize edge capacities as flows within the directed graph are interdependent; upstream and downstream edges can restrict total flow, preventing the full utilization of higher-capacity edges due to bottlenecks or imbalanced capacities. For instance, a low capacity on the edge $(\phi_i^{out}, \delta_i^{in})$ can restrict the flow on edge $(\delta_i^{in}, \delta_i^{out})$ from reaching node capacity. Therefore, iteratively refining the directed graph is essential.

### 3.4 ITERATIVE REFINEMENT

§3.3 presented how we obtain the max flow for a given graph partition; now we introduce how we can iteratively refine the graph partition to maximize the flow. We refine the graph iteratively based on edge swapping, which is a common approach for optimizing graph partition (Hendrickson et al., 1995; Vaishali et al., 2018), and we further propose a *max-flow guided edge swap* operation, which uses max-flow outputs to guide the iterative refinement of the graph.

The *preflow-push algorithm* mentioned in §3.3 provides the detailed flow assignments necessary to analyze edge utilization (Waissi, 1994). By comparing the flow through each edge with its capacity, we can identify *bottleneck* and *underutilized* edges. Bottleneck edges are defined as those where the flow reaches capacity limits, preventing the directed graph from achieving a higher overall flow. Underutilized edges are those where the flow falls short of capacity and could accommodate more data flow. *As long as these imbalances exist, we attempt to swap edges.* Therefore, we implement local swaps of edges guided by the max-flow outputs to form a new graph partition, as illustrated in Figure 3. This swap operation is essential in terms of: ([i]) balancing the inter- and intra-group edge weights to maintain high intra-group capacities while enabling efficient inter-group KV cache communicating; and ([ii]) adjusting the node and edge weights across intra-groups to optimize resource allocation. After the swaps, we rerun the two-phase algorithm to obtain the optimal model placement strategy and max flow of the new graph partition. We then refine the partition again. This iterative process continues until no further improvements can be made. Evaluation in §5.3 highlights the necessity of our design, the max flow guided edge swap overcomes local minima and accelerates optimization compared with other approaches. To better illustrate each phase of our scheduling algorithm, we provide a detailed analysis in Appendix C, and a case study in Appendix E.

## 4 SYSTEM IMPLEMENTATION

HEXGEN-2 is a distributed system designed to support efficient LLM inference service under the disaggregated paradigm in heterogeneous environments. HEXGEN-2 uses a *task coordinator* to handle the dispatch of incoming LLM inference requests, which is based on an open-source im-

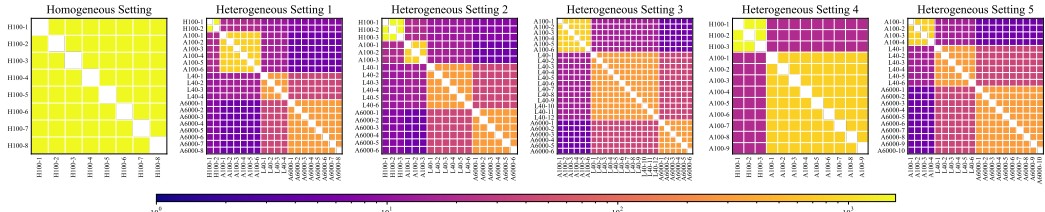

Figure 4: Communication bandwidth (Gbps) matrix for different settings. Homogeneous setting contains 8×H100 GPUs with a budget of 29.5 \$/$h$; heterogeneous setting 1 contains 2×H100, 6×A100, 4×L40 and 8×A6000 GPUs with a budget of 28.8 \$/$h$; heterogeneous setting 2 contains 3×H100 and A100, 6×L40 and A6000 GPUs with a budget of 26.9 \$/$h$; heterogeneous setting 3 contains 6×A100 and A6000, 12×L40 GPUs with a budget of 27.1 \$/$h$; heterogeneous setting 4 contains 3×H100 and 9×A100 GPUs with a budget of 26.3 \$/$h$; heterogeneous setting 5 contains 4×A100, 6×L40 and 10×A6000 with a 70% budget of 20.5 \$/$h$.

plementation of decentralized computation coordination (Yao, 2023) that utilizes libP2P (LibP2P, 2023) to establish connections among the work groups in a peer-to-peer network. All parallel communications in HEXGEN-2 are implemented using NVIDIA Collective Communication Library (NCCL) (NVIDIA, 2024), and all required communication groups for different parallelism plans are established in advance to avoid the overhead associated with constructing NCCL groups. HEXGEN-2 utilizes asynchronous NCCL `SendRecv`/`CudaMemcpy` for KV cache communication to enable overlapping between computation and communication. Furthermore, HEXGEN-2 integrates popular features for optimizing LLM inference such as continuous batching (Yu et al., 2022), FlashAttention (Dao et al., 2022; Dao, 2024), PagedAttention (Kwon et al., 2023), and supports open-source LLMs such as OPT (Zhang et al., 2022) and LLAMA (Touvron et al., 2023).

## 5  EVALUATION

To evaluate the design and implementation of HEXGEN-2, we ask the following essential questions:

- *What is the end-to-end performance comparison in terms of throughput and latency between* HEXGEN-2 *and the state-of-the-art homogeneous or heterogeneous generative inference systems?*

- *How effective is our scheduling algorithm in terms of finding the optimal assignment of the inference workflow compared with existing methods?*

### 5.1  EXPERIMENTAL SETUP

**Distributed environment.** We rent GPUs from RunPod (RunPod, 2023), a GPU cloud provider with services for various GPUs, and perform evaluation in the following setups:

- **Homogeneous setup:** We rent one on-demand instance equipped with 8×NVIDIA H100-80G GPUs, with a budget of \$29.52/hour to represent the standard homogeneous case.

- **Heterogeneous setups:** We utilize four types of GPUs: H100, A100, L40, and A6000, to construct five different heterogeneous cluster setups, where the first four settings use a similar budget as the homogeneous setting, while the last setting use a 70% budget of the homogeneous settings. The detailed configuration is illustrated in Figure 4.

We measure the communication bandwidth between each pair of GPUs via NCCL for all above mentioned environments. As shown in Figure 4, the heterogeneous environments demonstrate notable bandwidth limitation and heterogeneity.

**LLM inference workloads.** To evaluate the performances in different LLM inference workloads, we run four different types of workloads: heavy prefill with light decoding (HPLD), heavy prefill with heavy decoding (HPHD), light prefill with heavy decoding (LPHD), light prefill with light decoding (LPLD). Prefill requests that have more than 512 tokens are categorized as heavy, others are light, and decoding requests with more than 128 tokens are categorized as heavy (Hu et al., 2024). We generate these workloads using samples from the Azure Conversation dataset (Patel et al., 2024).

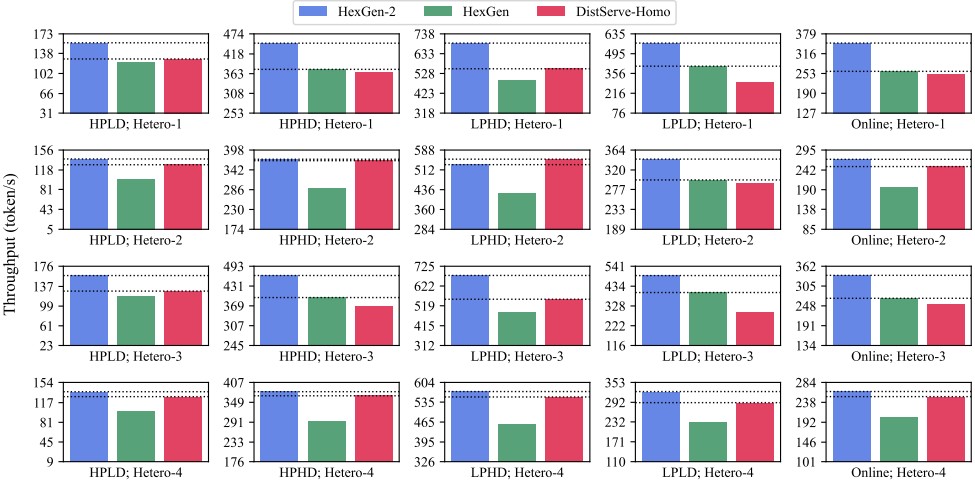

Figure 6: Throughput results to evaluate HEXGEN-2 on LLAMA-2 (70B). Each row corresponds to a particular heterogeneous setting. The first four columns demonstrates the offline inference results on different LLM workloads. The last column represents the online inference results.

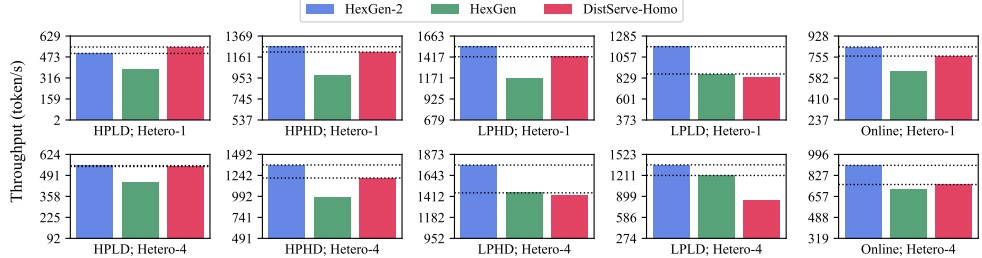

Figure 7: Throughput results to evaluate HEXGEN-2 on OPT (30B).

**Online and offline testing.** We test two different arrival rates: In the *online setting*, we scale the average arrival rate to 75% of the cluster's peak throughput to prevent request bursts that could cause system outages due to out-of-memory (OOM) errors, Figure 5 illustrates the distribution of input and output lengths in our trace. In the *offline setting*, we permit requests to arrive at a rate that fully utilizes the cluster, testing all four types of workloads (HPLD, HPHD, LPHD, LPLD).

**Models.** We evaluate HEXGEN-2 on OPT (30B) (Zhang et al., 2022) and LLAMA-2 (70B) (Touvron et al., 2023) models, both are representative and popular open-source transformer models, to study the system performance on models of different sizes.

**Baselines.** We carefully select state-of-the-art approaches as baselines. To understand end-to-end performance, we compare HEXGEN-2 with DIST-SERVE (Zhong et al., 2024) as the state-of-the-art

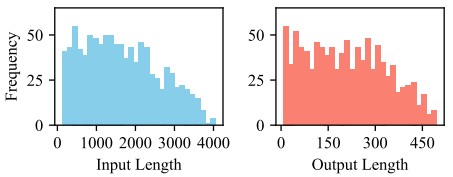

Figure 5: Request traces for online testing.

approach under the homogeneous setting, which enhances LLM serving by disaggregating prefill and decoding computations across different GPUs, allowing different resource allocation and parallelism for each phase. And HEXGEN (Jiang et al., 2024b) as the state-of-the-art approach under heterogeneous settings, which is a distributed inference engine that efficiently manages LLM inference across heterogeneous environments, leveraging asymmetric parallelism with a scheduling algorithm to optimize resource allocation. To understand the efficiency of the proposed scheduling algorithm, we compare its convergence with the truncated variant of our scheduling algorithm and *genetic algorithm*.

**Evaluation metrics.** For offline serving, we report the average decoding throughput, measured as the number of tokens generated per second. For online serving, we additionally report the SLO attainments as detailed in §2.

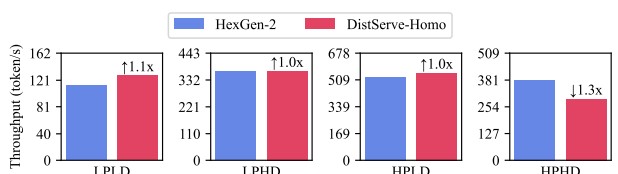

Figure 8: Latency results in online experiments.

## 5.2 END-TO-END EXPERIMENTAL RESULTS

**End-to-end performances.** Figure 6 and Figure 7 demonstrate the end-to-end throughput results of HEXGEN-2 compared with HEXGEN with different models, workloads, and heterogeneous settings, and DISTSERVE in the homogeneous setting. Given the same price budget, HEXGEN-2 outperforms its counterparts in almost all cases. In fact, compared with HEXGEN, HEXGEN-2 achieves up to a $1.5\times$ and, on average, a $1.4\times$ increase in serving throughput. Compared with DISTSERVE, HEXGEN-2 achieves up to a $2\times$ and, on average, a $1.3\times$ higher serving throughput. We also demonstrate the latency results of HEXGEN-2 compared with HEXGEN in different heterogeneous settings and with DISTSERVE in the homogeneous setting. As shown in Figure 8, HEXGEN-2 achieves on average a $1.5\times$ lower latency deadlines than its counterparts. Specifically, analyzing the scheduling results[2] under different heterogeneous settings and LLM workloads, we find that: (1) our scheduling approach prioritizes tensor model parallelism for prefill model replica to minimize latency and hybrid parallelism for decoding model replica to maximize throughput; (2) the scheduled result also employs pipeline parallelism to reduce the inter-machine communication over limited bandwidth, and avoid ultra-low cross data center communication; (3) relatively more resources are assigned for prefill and decoding in the HPLD and LPHD workloads to balance the resource demands for different phases; (4) our approach always schedules KV cache communications through high-bandwidth links such as NVLink and PCIe to prevent them from becoming system bottlenecks. We also compare HEXGEN-2 with the state-of-the-art LLM serving platform vLLM in Appendix F, and demonstrate the performance of HEXGEN-2 in the homogeneous setup in Appendix G.

**Cost efficiency.** To evaluate cost-efficiency in terms of serving throughput between homogeneous and heterogeneous setup, we reduce the budget in the heterogeneous setting by $30\%$. As shown in Figure 9, HEXGEN-2 in heterogeneous setting 5 still reveals similar performance

Figure 9: Throughput results with 70% price budget.

to DISTSERVE in the homogeneous setting, and even outperforms it by $30\%$ in some specific workloads. We believe that this is strong evidence to illustrate that a heterogeneous system such as HEXGEN-2 is capable of managing heterogeneous GPUs to provide more economical LLM inference services without compromising service quality.

## 5.3 EFFECTIVENESS OF THE SCHEDULING ALGORITHM

To evaluate the effectiveness of our scheduling algorithm, we compared its convergence behavior with some truncated variants, which disables the max-flow guided edge swap operation mentioned in §3.4 by replacing it with a random swap operation, and with the genetic algorithm. The genetic algorithm, designed to optimize model deployment, uses a population-based approach involving merge, split, and swap operations to iteratively refine GPU groupings (Jiang et al., 2024b). In our comparison, we replaced the group generation step in the graph partition phase and the iterative refinement phases of our algorithm with the genetic algorithm to enable HEXGEN-2 with this method. We benchmarked heterogeneous setting 1 across all four types of workloads. Figure 10 and Figure 11 illustrate the convergence curves and experimental results. Our scheduling algorithm identifies optimal assignments for all scenarios within 90 to 120 seconds, which significantly outperforms both the truncated variant and the genetic algorithm, finds assignments that deliver on average a $1.8\times$ higher serving throughput and converges much faster, while the others get stuck in local minima. Additionally, we verified that in all cases, the estimated serving throughput closely aligns with the actual throughput. Our scheduling algorithm also scales effectively with larger clusters, we demonstrate the experimental results in Appendix H.

---

[2]The placements chosen by HEXGEN-2 for online experiments can be found in Appendix B.

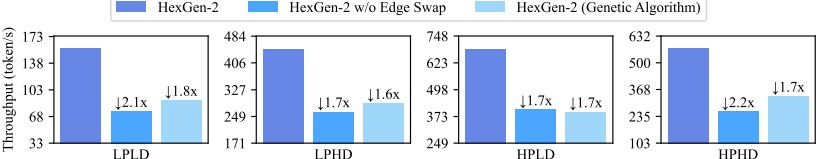

Figure 10: Convergence comparison of our proposed search strategy, our strategy without edge swap, and genetic algorithm, where all run 15 times.

Figure 11: Throughput comparison in heterogeneous setting 1 among HEXGEN-2, HEXGEN-2 without edge swap, and HEXGEN-2 empowered by genetic algorithm.

## 6  RELATED WORKS

**LLM inference serving and disaggregated inference paradigm.** There are plenty of recent researches focused on optimizing LLM inference and serving  (Li et al., 2023; Kwon et al., 2023; Agrawal et al., 2024; Liu et al., 2023; Wu et al., 2023; Zhou et al., 2022; Yu et al., 2022). Among them, vLLM (Kwon et al., 2023) proposes paged-attention to improve the memory efficiency of the system. Orca (Yu et al., 2022) introduces continuous batching to improve inference throughput. AlpaServe (Li et al., 2023) adopts model parallelism to optimize LLM serving performance. SARATHI (Agrawal et al., 2024) introduces a chunked-prefill approach and piggybacks decoding requests to improve hardware utilization. Deja Vu (Liu et al., 2023) predicts contextual sparsity on-the-fly and uses an asynchronous and hardware-aware implementation to enhance LLM inference. On the other hand, many very recent works have been produced using disaggregated paradigm. Splitwise (Patel et al., 2024) splits the prefill and decoding phases onto separate machines to optimize hardware utilization. DistServe (Zhong et al., 2024) further implements distinct parallel strategies for different phases. TetriInfer (Hu et al., 2024) partitions prompts into fixed-size chunks and adopts a two-level scheduling algorithm to improve the performance of disaggregated inference. Mooncake (Qin et al., 2024) features a KV cache-centric disaggregated architecture that enhances inference by fully leveraging the underutilized resources of GPU clusters, excelling in long-context scenarios. These works further confirm the effectiveness of the disaggregated architecture.

**Heterogeneous GPU computing.** Recent efforts have investigated diverse approaches to deploying LLMs in heterogeneous environments. LLM-PQ (Zhao et al., 2024) supports adaptive model quantization and phase-aware partitioning to boost LLM serving efficiency on heterogeneous GPU clusters. Helix (Mei et al., 2024) formulates heterogeneous GPUs and network connections as a maxflow problem, and adopts a mixed integer linear programming algorithm to discover highly optimized strategies for serving LLMs. HexGen (Jiang et al., 2024b) proposes asymmetric parallelism and an advanced scheduling algorithm to deploy generative inference in decentralized and heterogeneous environments. Mélange (Griggs et al., 2024) formulates the GPU allocation task as a cost-aware bin packing problem and optimizes cost efficiency for LLM services by leveraging heterogeneous GPU types. Note that our work shares a similar objective and but is the first to adapt the disaggregated inference architecture for heterogeneous environments.

## 7  CONCLUSION

We explore the potential of implementing a disaggregated inference framework in heterogeneous environments with devices of diversified computational capacities connected over a heterogeneous network. Toward this end, we propose HEXGEN-2, a generative inference framework that incorporates a disaggregated architecture alongside an efficient scheduling algorithm tailored for such deployments. Our empirical study suggests that, given the same budget, HEXGEN-2 can outperform state-of-the-art homogeneous and heterogeneous inference frameworks by up to $2.0\times$ and on average $1.3\times$ in serving throughput, and reduces the average inference latency by $1.5\times$. Additionally, HEXGEN-2 maintains competitive inference performance relative to leading frameworks with a $30\%$ lower price budget. We believe that such an effort from HEXGEN-2 to provide *efficient economical* LLM inference could potentially democratize the usage of generative AI.

ACKNOWLEDGMENT

This work is supported by the HKUST startup grant R9895 from CSE; RGC-ECS project 26218024; RGC-NSFC project CRS_HKUST601/24; HKUST-WeBank Joint Lab project WEB24EG01-E; HKUST-Huawei Joint Lab project 24250360Y043.

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

Table 1: Modeling the generative inference cost and limit.

| Description | Prefill Cost Formulation | Decode Cost Formulation |
|---|---|---|
| Computation cost | $\max_{d \in \mathbf{d}_{i,j}} \left( \frac{24 b_t s_t^{\text{in}} H^2}{\|\mathbf{d}_{i,j}\| c_d} \right) \cdot l_{i,j}$ | $\max_{d \in \mathbf{d}_{i,j}} \left( \frac{12 H^2 B_{\text{type}} s_t^{\text{out}}}{\|\mathbf{d}_{i,j}\| m_d} \right) \cdot l_{i,j} + \max_{d \in \mathbf{d}_{i,j}} \left( \frac{24 b_t s_t^{\text{out}} H^2}{\|\mathbf{d}_{i,j}\| c_d} \right) \cdot l_{i,j}$ |
| TP communication cost | $\max_{d \in \mathbf{d}_{i,j}} \left( \sum_{d' \in \mathbf{d}_{i,j} \setminus \{d\}} \left( \alpha_{d,d'} + \frac{b_t s_t^{\text{in}} H B_{\text{type}}}{\|\mathbf{d}_{i,j}\| \beta_{d,d'}} \right) \right) \cdot 4 l_{i,j}$ | $\max_{d \in \mathbf{d}_{i,j}} \left( \sum_{d' \in \mathbf{d}_{i,j} \setminus \{d\}} \left( \alpha_{d,d'} + \frac{b_t H B_{\text{type}}}{\|\mathbf{d}_{i,j}\| \beta_{d,d'}} \right) \right) \cdot 4 s_t^{\text{out}} l_{i,j}$ |
| PP communication cost | $\min_{d \in \mathbf{d}_{i,j}, d' \in \mathbf{d}_{i,j+1}} \left( \alpha_{d,d'} + \frac{b_t s_t^{\text{in}} H B_{\text{type}}}{\beta_{d,d'}} \right)$ | $\min_{d \in \mathbf{d}_{i,j}, d' \in \mathbf{d}_{i,j+1}} \left( \alpha_{d,d'} + \frac{b_t H B_{\text{type}}}{\beta_{d,d'}} \right) \cdot s_t^{\text{out}}$ |
| Memory limit | $\left( \frac{12 H^2 B_{\text{type}}}{\|\mathbf{d}_{i,j}\|} + \frac{2 b_t \left( s_t^{\text{in}} + s_t^{\text{out}} \right) H B_{\text{type}}}{\|\mathbf{d}_{i,j}\|} \right) \times l_{i,j} + \quad 4 b_t \left( s_t^{\text{in}} + s_t^{\text{out}} \right) H B_{\text{type}}$ | |
| KV cache communication cost | $\alpha_{d,d'} + \frac{2 b_t s_t^{\text{in}} H B_{\text{type}}}{\beta_{d,d'}}$ | |

We formulate the computation cost, tensor parallel (TP) communication cost, key-value (KV) cache communication cost, memory limit of the $j$-th stage in the $i$-th pipeline, and the pipeline parallel (PP) communication cost between the $j$-th and $(j{+}1)$-th stages of the $i$-th pipeline for a particular inference task $t \in \mathbf{T}$. Here, $d$ is the GPU device, $m_d$ is the GPU memory bandwidth, $c_d$ is the tensor core computation power, $\alpha_{d,d'}$ and $\beta_{d,d'}$ is the latency and bandwidth between device $d$ and $d'$, $\mathbf{d}_{i,j}$ is the set of GPUs serves the $j$-th stage in the $i$-th pipeline that holds $l_{i,j}$ transformer layers, $b_t$ is the batch size, $s_t^{\text{in}}$ is the sequence length of the input prompt, $s_t^{\text{out}}$ is the number of output tokens, $H$ is the size of the hidden dimension in a transformer block, and $B_{\text{type}}$ denotes the number of bytes for the precision of inference computation (e.g., $B_{\text{type}}(\text{FP16}) = 2$).

Table 2: GPU Deployment, Strategy, and Type.

| LLAMA-2 (70B) | | | | | |
|---|---|---|---|---|---|
| **Heterogeneous Setting 1** | | | **Heterogeneous Setting 3** | | |
| GPU Configuration | Strategy | Type of Instance | GPU Configuration | Strategy | Type of Instance |
| 1xH100+1xA100 | TP=1,PP=2 | Prefill Instance | 2xA100 | TP=1,PP=2 | Prefill Instance |
| 2xA100+2xA6000 | TP=2,PP=2 | Prefill Instance | 2xL40+3xA6000 | TP=1,PP=5 | Prefill Instance |
| 4xL40 | TP=4,PP=1 | Prefill Instance | 4xL40 | TP=4,PP=1 | Prefill Instance |
| 1xH100+1xA100 | TP=1,PP=2 | Decode Instance | 4xA100 | TP=2,PP=2 | Decode Instance |
| 2xA100+2xA6000 | TP=2,PP=2 | Decode Instance | 2xL40+3xA6000 | TP=1,PP=5 | Decode Instance |
| 4xL40 | TP=2,PP=2 | Decode Instance | 4xL40 | TP=2,PP=2 | Decode Instance |
| **Heterogeneous Setting 2** | | | **Heterogeneous Setting 4** | | |
| GPU Configuration | Strategy | Type of Instance | GPU Configuration | Strategy | Type of Instance |
| 1xH100+1xA100 | TP=1,PP=2 | Prefill Instance | 1xH100+1xA100 | TP=1,PP=2 | Prefill Instance |
| 2xL40+2xA6000 | TP=2,PP=2 | Prefill Instance | 2xA100 | TP=2,PP=1 | Prefill Instance |
| 2xH100+2xA100 | TP=2,PP=2 | Decode Instance | 2xH100+2xA100 | TP=2,PP=2 | Decode Instance |
| 4xL40+4xA6000 | TP=4,PP=2 | Decode Instance | 4xA100 | TP=4,PP=1 | Decode Instance |
| OPT (30B) | | | | | |
| **Heterogeneous Setting 1** | | | **Heterogeneous Setting 4** | | |
| GPU Configuration | Strategy | Type of Instance | GPU Configuration | Strategy | Type of Instance |
| 1xH100+1xA100 | TP=1,PP=2 | Prefill Instance | 1xH100 | TP=1,PP=1 | Prefill Instance |
| 2xA100 | TP=2,PP=1 | Prefill Instance | 1xA100 | TP=1,PP=1 | Prefill Instance |
| 2xL40+1xA6000 | TP=1,PP=3 | Prefill Instance | 1xA100 | TP=1,PP=1 | Prefill Instance |
| 2xL40+1xA6000 | TP=1,PP=3 | Prefill Instance | 1xA100 | TP=1,PP=1 | Prefill Instance |
| 1xH100+1xA100 | TP=1,PP=2 | Decode Instance | 2xH100 | TP=2,PP=1 | Decode Instance |
| 2xA100 | TP=1,PP=2 | Decode Instance | 2xA100 | TP=1,PP=2 | Decode Instance |
| 2xL40+1xA6000 | TP=1,PP=3 | Decode Instance | 2xA100 | TP=1,PP=2 | Decode Instance |
| 2xL40+1xA6000 | TP=1,PP=3 | Decode Instance | 2xA100 | TP=1,PP=2 | Decode Instance |

# A   GENERATIVE INFERENCE COST ESTIMATION

**Node capacity estimation.** To estimate the generative inference cost, we adopt the *cost model* from HEXGEN (Jiang et al., 2024b) and summarize the computation costs, communication costs, and memory consumption constraints in Table 1. The inference latency for a single request is calculated by summing the total computation and communication costs. We determine the capacity of the compute-bound prefill node, where batching more requests does not enhance system throughput, by dividing the predefined time period by the latency. Conversely, for the memory I/O-bound decoding node, which benefits from batching, we calculate its capacity by dividing the product of the maximum available batch size and the time period by the latency.

**Edge capacity estimation.** For connection types (1) and (2) mentioned in §3.3, the edge capacities are equal to the product of the predefined time period and the connection bandwidth, divided by the transmission size of a request. For connection type (3), the edge capacity is equal to the time period divided by the estimated KV cache communication cost in Table 1. As mentioned in §3.3, the edge capacity of connection type (3) is determined by the collective performance of all GPU-to-GPU transmission connections, as each connection is responsible for a portion of the KV cache transmission. To optimize it, given the parallel configurations of the prefill and decoding model replicas,

we adjust the pipeline stage order of both phases to minimize the overall KV cache communication cost, which in turn determines the edge capacity.

**Memory requirement estimation for a single model replica.** The memory cost model in Table 1 estimates the memory required for a single model replica. To determine the total memory requirement for a single model replica, we assume a batch size of 32 concurrent requests (i.e., $b_t = 32$). Thus, the total memory requirement is calculated as: model parameter size + 32 * KV cache size per request.

## B   HEXGEN-2 SCHEDULING RESULTS

We list the model serving group partitions and types generated by HEXGEN-2 in the online experiments for each heterogeneous setting in Table 2.

## C   SCHEDULING ALGORITHM ANALYSIS

The scheduling algorithm aims to optimize the deployment of large language model (LLM) inference workloads on a heterogeneous GPU cluster. The optimization involves the following essential phases:

- **Graph partition.** The initial partition focuses on creating memory-balanced groups and optimizing the capacity within each group. The secondary partition determines group type (i.e., prefill or decoding), focusing on maximizing inter-type communication bandwidth for efficient KV cache transfer.

- **Max-flow.** This phase determines optimal parallel strategies for each group and determines the optimal inter-type KV cache communication paths based on the max-flow outputs.

- **Iterative refinement.** This phase continuously adjusts partitions and strategies based on workload demands until no further improvements can be made.

**The upper bound for graph partitioning** indicates *the optimal utilization of heterogeneous computation power and connections.* The theoretical upper bound of the graph partition phase is achieved when the cluster is partitioned into groups with balanced memory capacities and optimized processing capabilities, and the groups are assigned types (i.e., prefill or decoding) in a manner that maximizes inter-type communication bandwidth for key-value (KV) cache transfers.

**The upper bound for max-flow** indicates *the maximum possible data flow within the cluster.* The theoretical upper bound of the max flow phase is determined by the maximum possible data transfer rate of the entire system. This upper limit is achieved when the system fully utilizes the inter-type network bandwidth for KV cache transfers and optimizes the processing capabilities of the prefill and decoding model replicas.

Based on our scheduling algorithm, the optimization will iteratively narrow the gap between the current allocation and the theoretical upper bounds, where the iterative refinement process *addresses the limitations inherent in each phase.* The challenges in reaching upper bounds lie in two aspects:

- **In the graph partition phase,** creating an ideal graph partition in a single iteration is challenging since this phase lacks critical information (e.g., parallel strategy and KV cache communication path) from subsequent phases. Without these insights, the initial graph partitioning cannot guarantee an ideal utilization of the heterogeneous cluster, leading to potential communication bottlenecks and workload imbalances.

- **The max flow phase** operates within the constraints set by the graph partition. The max-flow algorithm cannot achieve the theoretical maximum flow if the preceding graph partition results in less-than-optimal grouping. Limited inter-group communication bandwidth and unbalanced node capacities prevent the system from fully utilizing the network's data transfer capabilities.

**Iterative refinement.** *The iterative refinement phase is crucial in bridging the gap toward the upper bounds.* It continuously evaluates and adjusts groupings, fine-tunes parallel configurations and recalculates optimal KV cache communication paths based on updated partitions. This approach allows the algorithm to:

- **Rebalance trade-offs for graph partition.** Balance intra-group resource optimization with inter-type communication efficiency for optimized resource utilization.

- **Enhance max-flow potential.** Balance overutilized and underutilized edges within the formulated flow network for optimized data flow efficiency.

# D    DISAGGREGATION AND CHUNKED PREFILL

Chunked prefill (Agrawal et al., 2024) is a method that divides input tokens into smaller chunks, which are then processed in a continuous batch. This approach simplifies scheduling by treating all nodes uniformly and enhances computational efficiency during decoding, improving machine utilization. However, this approach may not result in significant performance gains across all workload types. We evaluate chunked prefill using vLLM (Kwon et al., 2023) on one H100 GPU serving the OPT-30B model. Experimental results demonstrate that on HPLD and LPLD workloads, chunked prefill brings an approximately 20% throughput improvement, while it only brings around 5% throughput gains on HPHD and LPHD workloads. Therefore, we choose disaggregation, which enables different batching strategies, resource allocations, and parallel approaches for each phase, providing greater flexibility in handling various types of workloads.

# E    CASE STUDY: SCHEDULING ALGORITHM ANALYSIS ON A SMALL CLUSTER

In this section, we provide a case study of our scheduling algorithm on relatively small size heterogeneous cluster with 4 H100s and 4 A100s for better understanding of our scheduling algorithm. The detailed procedures are listed bellow.

## E.1    PHASE 1: GRAPH PARTITION

The graph partition phase aims to find the group construction and type mentioned in §3.1.

**Step 1: initial partition.** Step 1 divides the GPUs into multiple independent groups based on minimizing inter-group communication bandwidth and balancing the memory capacity of each group. After step 1, the cluster is divided into four groups g1-4, and the construction of each group is: g1: two H100, g2: two H100, g3: two A100, and g4: two A100.

**Step 2 & 3: coarsen & secondary partition & projection.** This step aims to distinguish the type for each group (prefill or decoding). In the small case, g1 and g3 are determined to be the prefill model replicas, and g2 and g4 are determined to be the decoding model replicas.

## E.2    PHASE 2: MAX-FLOW ALGORITHM

The max-flow algorithm aims to fine the parallel strategy and KV cache communication path mentioned in §3.1.

**Step 1: find the optimal parallel strategies for prefill and decoding groups.** This step determines the latency- and throughput-optimal parallel strategies for prefill and decoding model replicas. After searching, g1 and g3 (prefill model replicas) use a parallel strategy of (TP=2, PP=1) (latency-optimal), while g2 and g4 (decoding model replicas) use a parallel strategy of (TP=1, PP=2) (throughput-optimal).

**Step 2: find the optimal KV communication path.** We run a preflow-push algorithm to get the max flow of the cluster. The generated flow assignments are used to guide the KV cache communication. In the small case, g1 (prefill model replica) communicates with g2 (decoding model replica), and g3 (prefill model replica) communicates with g4 (decoding model replica).

### E.3 PHASE 3: ITERATIVE REFINEMENT

The iterative refinement phase aims at co-optimizes the four objectives (group construction, group type, parallel strategy and KV cache communication path) in the first and second phases.

**Iterative refinement using swap operation.** We use max-flow guided edge swap to iterative refine the graph partition until no further improvements can be made. For instance, for workloads with light prefill and heavy decoding (LPHD) needs, the algorithm would attempt to allocate more resources to decoding model replicas. In the small case with LPHD workloads, one H100 from g1 (prefill model replica) is swapped into g2 (decoding model replica) and one A100 from g3 (prefill model replica) is swapped into g4 (decoding model replica) for enhancing the decoding ability of the system and maximizing the system throughput. The iterative refinement will optimize the plan for any given LLM inference workload accordingly given the workload characteristics.

In this small case, the output of our scheduling algorithm is the same as the output that is derived through exhaustive search.

## F COMPARE HEXGEN-2 WITH VLLM

In this section, we conduct additional experiments to compare HEXGEN-2 with state-of-the-art LLM serving platform. We evaluated vLLM using the same homogeneous experimental setup described in §5.1. Specifically, we rent 8 H100 GPUs from the RunPod platform and test vLLM with the Llama2-70B model using samples from the Azure Conversation dataset. As demonstrated in Table 3, HEXGEN-2 achieves up to a $2.1\times$ and on average a $1.5\times$ higher serving throughput compared with vLLM in our testbed.

Table 3: Comparison between different frameworks with different setups.

| Setting | System | HPLD | HPHD | LPHD | LPLD | Online |
|---|---|---|---|---|---|---|
| Heterogeneous Setting 1 | HEXGEN-2 | 157 tokens/s | 448 tokens/s | 689 tokens/s | 570 tokens/s | 350 tokens/s |
| Heterogeneous Setting 1 | HEXGEN | 123 tokens/s | 375 tokens/s | 492 tokens/s | 407 tokens/s | 259 tokens/s |
| Homogeneous Setting | DISTSERVE | 128 tokens/s | 368 tokens/s | 553 tokens/s | 291 tokens/s | 251 tokens/s |
| Homogeneous Setting | VLLM | 97 tokens/s | 437 tokens/s | 563 tokens/s | 270 tokens/s | 256 tokens/s |

## G CASE STUDY: HOMOGENEOUS SYSTEM COMPARISON

In this section, we compare HEXGEN-2 with DISTSERVE and HEXGEN in a homogeneous setup.

**Experimental setup.** To compare the runtime of HEXGEN-2 with DISTSERVE and HEXGEN, we rented 4 H100 GPUs from the RunPod platform and tested serving throughput on the OPT-30B model using the four types of LLM inference workloads (HPLD, HPHD, LPHD, LPLD) described in §5.1.

**Compare with DISTSERVE.** We found that for certain inference workloads, the scheduling results of HEXGEN-2 and DISTSERVE differ. For example, with the HPLD workload, HEXGEN-2 favors replicating more model replicas to enhance the system's parallel processing, while DISTSERVE prefers model parallelism to distribute the computation of a single model replica across multiple GPUs. Experimental results demonstrate that HEXGEN-2 outperforms DISTSERVE in certain cases due to better scheduling results while delivering comparable performance when the scheduling outcomes are the same.

**Compare with HEXGEN.** HEXGEN-2, with optimized scheduling in a disaggregated architecture, minimizes interference between the prefill and decoding phases of LLM inference. It selects appropriate parallelism and batching strategies for each phase, resulting in improved inference performance compared to HEXGEN in a homogeneous environment.

## H CASE STUDY: SCHEDULING ALGORITHM SCALABILITY

In this section, we conduct additional experiments to evaluate the scalability of our scheduling algorithm. The results are shown below.

Table 4: Throughput comparison in a homogeneous cluster.

|        | HEXGEN-2     | DISTSERVE    | HEXGEN       |
|--------|--------------|--------------|--------------|
| HPLD   | 365 tokens/s | 302 tokens/s | 277 tokens/s |
| HPHD   | 683 tokens/s | 692 tokens/s | 505 tokens/s |
| LPHD   | 758 tokens/s | 774 tokens/s | 533 tokens/s |
| LPLD   | 730 tokens/s | 553 tokens/s | 545 tokens/s |

Table 5: Algorithm convergence time across different cluster sizes.

| Ngpus | Time (min) |
|-------|------------|
| 64    | 4.03       |
| 128   | 7.93       |
| 192   | 21.66      |
| 256   | 28.44      |
| 320   | 47.77      |

Experimental results demonstrate that our scheduling algorithm scales polynomially and shows potential for addressing larger and more complex heterogeneous scheduling problems.

