# OpenReview forum: "HexGen-2: Disaggregated Generative Inference of LLMs in Heterogeneous Environment"
_ICLR.cc/2025/Conference — ICLR 2025 Poster_

### Official Review · Reviewer_Rq11 · 2024-11-01

**Soundness:** 3
**Presentation:** 2
**Contribution:** 3
**Rating:** 5
**Confidence:** 3

**Summary:**

Traditional LLM serving frameworks co-locate the execution of prefill and decode stages, leading to prefill-decode interference.
Disaggregated inference is a more efficient LLM serving approach that reduces such interference and allows more flexible scheduling of the two stages.
This paper proposes HexGen-2, a disaggregated LLM inference framework that coordinates distributed LLM inference over a heterogeneous set of GPUs and network connections.
The core of HexGen-2 is a two-level scheduling algorithm that leverages graph partitioning and max-flow algorithm to determine the resource allocation and parallelization plan for the prefill and decode stage.
Evaluations have shown that HexGen-2 could achieve better serving throughput and lower average latency compared to SOTA works.

**Strengths:**

1. The problem of scheduling heterogeneous resources over disaggregated LLM inference makes sense.
2. The framework shows good improvement over prior works on serving throughput and latency.

**Weaknesses:**

1. Lack of intuition and lack of clear and detailed explanation for the core scheduling algorithm.
2. Did not compare the scheduling decision with the optimal solution in small cases.
3. Lack of justification on scalability of the framework.

**Questions:**

Thank you for submitting to ICLR 2025! I think this paper tries to tackle the important problem of scheduling heterogeneous GPU and network resources for disaggregated LLM serving.
Despite the good evaluation results, I have a few comments for the paper and it would be great if the authors could address them.

The current explanation for the core scheduling algorithm is confusing and unclear.
There is no clear intuition on why the algorithm should be designed at these two levels.
There is also no clear guidance on what the objective is during each phase of the algorithm.
For example, in the first step of the first phase: graph partition, is balancing the node weights (memory capacity) the overall objective at this step?
The node weight in the global graph seems to be determined by GPU memory but not GPU compute, why is that the case?
Why is the coarsen step necessary in Step 2?
My understanding is that a prefill or decode replica could have multiple super nodes, is that right?

In addition, in the second phase, why is it required that each compute node needs to be connected to two other nodes in the same graph?
How are latency-optimal configuration and throughput-optimal configuration for prefill and decode replicas respectively related to the max-flow algorithm used?
In iterative refinement, what is the physical meaning of swapping edges?
I think the scheduling algorithm should also have easy-to-follow examples on the side to clearly give intuitions to readers.
The current Figure 3 is hard to understand even after reading the entire Section 3.

I am also wondering how the current algorithm compares to the optimal allocation and parallelization plan.
Despite the NP-hardness of the problem, the optimal plan should be solvable by just brutal forcing all possible plans in small cases such as 4 H100 and 4 A100 GPUs.
In Section 5.3, it says "Our scheduling algorithm identifies optimal assignments for all scenarios within 90 to 120 seconds".
How is optimality defined here?
Is the algorithm always guaranteed to find the optimal solution given enough search time?

Also, how much overhead would the algorithm incur if running on a cluster with, for example, hundreds of GPUs?
It may be hard to rent such a large number of GPUs for experiments, but evaluating algorithm overhead should be possible.

In the evaluation, it says HexGen-2 is compared against DistServe under the homogeneous setting with 8 H100 GPUs.
Why is DistServe appearing in all heterogeneous settings in both Figure 6 and Figure 7?

---

> ### Author Response · Authors · 2024-11-21
>
> W1. Lack of intuition and lack of clear and detailed explanation for the core scheduling algorithm.
>
> Sorry for the unclear demonstration of the scheduling algorithm. We enumerate the reply for each of the mentioned question below:
>
> Q1.1. There is no clear intuition on why the algorithm should be designed at these two levels.
>
> Thanks for sharing this comment, we believe this is an issue of presentation rather than algorithm design. We summarize the intuitions of why the algorithm should be designed at these two levels are listed as follows:
>
> - **Effective decomposition simplifies complex optimization problems.** When determining group partition, group type, parallel strategy, and KV cache communication in a heterogeneous cluster, the search space is extremely large, making it computationally infeasible to solve in one step. By decomposing the problem into smaller sub-problems, we significantly reduce computational complexity, making the optimization more manageable.
> - **Specialized methods improve performance and efficiency.** For each distinct optimization phase, we employ algorithms and heuristics specifically suited to the sub-task. For example, graph partitioning is used for group partitioning and type determination, while max flow is used for determining optimal parallel and KV cache communication strategies. This targeted approach leads to faster convergence, ensuring a more effective solution compared to tackling the entire problem at once.
>
> Q1.2. There is also no clear guidance on what the objective is during each phase of the algorithm.
>
> *The objective during each phase of the algorithm:*
>
> - **Graph partition phase.** The initial partition focuses on creating memory-balanced groups and optimizing the capacity within each group. The secondary partition determines group type (prefill or decoding), focusing on maximizing inter-type communication bandwidth for efficient KV cache transfer.
> - **Max-flow phase.** This phase selects the optimal parallel configuration for each group and determines the KV cache communication path for prefill and decoding replicas.
> - **Iterative refinement phase.** This phase co-optimizes the graph partition and max-flow phases by iteratively adjusting partitions, types, and strategies until no further improvements are possible.
>
> We provided a case study in W2 & Q3.1 for a detailed analysis of each step of our scheduling algorithm (a small case with 4 H100 and 4 A100 GPUs).
>
> Q1.3. In the first step of the first phase: graph partition, is balancing the node weights (memory capacity) the overall objective at this step?
>
> No, beyond balancing node weights, the graph partition also *minimizes inter-group edge weights*. Specifically, the graph partition reduces inter-group edge weights (i.e., inter-group communication bandwidth) to maximize intra-group communication efficiency, thereby enhancing the processing capability within each model serving group.
>
> Q1.4. Why we need to balance the memory capacity rather than compute capacity?
>
> *The main objective is to avoid OOM issues and provide a good starting point for further optimization (iterative refinement).* Balancing compute capacity rather than memory often causes OOM issues in heterogeneous clusters, hindering the algorithm's convergence. *Note that memory capacity may become imbalanced among groups to adapt to varying inference workloads in the iterative refinement phase.* Concretely, the memory balancing process occurs only during the initial partition step of the graph partitioning phase. During the iterative refinement phase, group memory capacity may become imbalanced. For example, with light prefill and heavy decoding workloads, more memory capacities are assigned to the decoding model replicas to balance the resource needs across different inference phases.
>
> We have integrated the discussion into our updated draft (Section 3.2).
>
> Q1.5. Why is the coarsen step necessary in Step 2?
>
> *Coarsen operation simplifies the graph and enables a more effective partition.* Concretely, in complex heterogeneous environments, the global graph is typically large and complex. Directly partitioning the global graph into multiple parts usually generates poor partitioning results [1]. The coarsen operation is used to simplify the global graph into smaller graphs, which makes the partitioning more effective and is a typical optimization in graph partition problems.
>
> [1] Hendrickson B, Leland R W. A Multi-Level Algorithm For Partitioning Graphs[J]. SC, 1995, 95(28): 1-14.
>
> Q1.6. Could a prefill or decode replica have multiple super nodes?
>
> *No, a single super node represents a single prefill or decoding model replica.* Concretely, a super node represents the coarsened version of a model serving group, with each group responsible for serving either a prefill or decoding replica.

---

> ### Author Response · Authors · 2024-11-21
>
> Q2.1. In the second phase, why is it required that each compute node needs to be connected to two other nodes in the same graph?
>
> One compute node in the second phase could be a prefill model replica or a decoding model replica.
>
> - **Prefill model replica.** If this compute node is a prefill model replica, then it should accept incoming requests from the source node, and pass the KV cache to the decoding model replica, so each prefill model replica should be connected to two other nodes (source node and decode model replica).
> - **Decode model replica.** If this compute node is a decoding morel replica, it should accept the KV cache from the prefill model replica and pass the output response to the sink node, so each decode model replica should be connected to two other nodes (prefill model replica and sink node).
>
> Thus, each compute node needs to be connected to two other nodes in the same graph.
>
> Q2.2. How are latency-optimal configuration and throughput-optimal configuration for prefill and decode replicas respectively related to the max-flow algorithm used?
>
> We want to clarify that *these configurations are used to optimize the end-to-end system performance scheduled by the max-flow algorithm.* One essential goal of the disaggregated paradigm is to reduce first token latency, so a latency-optimal configuration is chosen for prefill model replicas to minimize latency. Improving decoding throughput is another central focus of the disaggregated paradigm, which can be achieved by batching more requests in the decoding phase. Therefore, a throughput-optimal configuration is chosen for decoding replicas to maximize throughput. Consequently, these default configurations are applied to optimize overall system performance, guided by the max-flow algorithm.
>
> Q2.3. In iterative refinement, what is the physical meaning of swapping edges?
>
> We clarify the physical meaning of edge swapping. Swapping an edge indicates transferring an edge (communication condition) from intra- to inter-group, which corresponds to moving a GPU (and its associated edges) from one model serving group to another. This operation affects the composition of each model serving group and can lead to improved performance by balancing computational and communication loads. We provide a simple example here: consider two model serving groups, g1 and g2, with 4 GPUs in g1 and 2 GPUs in g2. Swapping an intra-group edge from g1 to inter-group means moving 1 GPU and its connections from g1 to g2, resulting in both g1 and g2 having 3 GPUs each.

---

> ### Author Response · Authors · 2024-11-21
>
> W2 & Q3.1. Case study on small cluster case: 4 H100 and 4 A100.
>
> Given the cluster consists of 4 H100 and 4 A100 GPUs, our scheduling algorithm’s procedures are as illustrated below:
>
> - **Phase 1 graph partition.** Phase 1 creates groups that optimize memory usage and capacity, and designates group types to maximize inter-type communication bandwidth.
>     - **Step 1 initial partition:** divide the GPUs into independent groups g1 through g4 based on minimizing inter group edge weights (minimizing inter group communicaion bandwidth). This step ensures each group is memory-balanced and has optimized capacity.
>     Groups: g1: 2 H100 GPUs, g2: 2 H100 GPUs, g3: 2 A100 GPUs, g4: 2 A100 GPUs.
>     - **Step 2 coarsening:** merge each group into a super node to simplify the graph. This step simplifies the global graph for efficient secondary partitioning.
>     Super nodes: s1 represents g1, s2 represents g2, s3 represents g3, s4 represents g4.
>     - **Step 3 secondary partition:** divide super nodes into two partitions based on maximizing inter-partition communication bandwidth, and assign group types. This step maximizes bandwidth for KV cache transfer between prefill and decoding replicas.
>     Partitions: p1: prefill model replicas (s1 and s3), p2: decoding model replicas (s2 and s4).
>     - **Step 4 projection:** revert super nodes back to their original groups. This step assigns specific roles to each group based on the partitioning.
>     Prefill model replicas: g1 and g3, decoding model replicas: g2 and g4.
> - **Phase 2 max-flow algorithm.** Phase 2 determines optimal parallel strategies for each group and establishes efficient KV cache communication paths.
>     - **Step 1 determine optimal parallel strategies:** assign parallel configurations to each group based on their role. This step optimizes the processing capability of each replica based on their type.
>     For prefill model replicas (g1, g3), latency optimal parallel configuration is assigned, which is Tensor Parallelism (TP) = 2, Pipeline Parallelism (PP) = 1.  For decoding model replicas (g2, g4), throughput optimal parallel configuration is assigned, which is TP = 1, PP = 2.
>     - **Step 2 determine optimal KV communication path:** use the preflow-push algorithm to optimize data flow, and route KV cache communication based on the generated flow assignments. This step determines the optimal KV cache transmission path to maximize system throughput.
>     KV cache communication paths: g1 (prefill) ↔ g2 (decoding), g3 (prefill) ↔ g4 (decoding).
>     - **Phase 3 iterative refinement.** Phase 3 continuously adjust partitions and strategies based on workload demands until no further improvements can be made. This phase balances prefill and decoding capabilities, as well as optimizes KV communication efficiency to enhance overall system performance for varying inference workloads. Assume the coming workload is Light Prefill and Heavy Decoding (LPHD), this phase reallocates more resources to the decoding model replicas to better handle the load: swap one H100 GPU from g1 (prefill) to g2 (decoding), swap one A100 GPU from g3 (prefill) to g4 (decoding). The KV communication paths remain unchanged.
>
> In this small case, the output of our scheduling algorithm is the same as the output that is derived through exhaustive search. Note that while the case study uses a small cluster for illustration, the algorithm is designed to scale to large, complex, and heterogeneous clusters.
>
> We have incorporated the case study into our updated paper (Appendix E) for better illustration of our problem, thank you for your suggestion.
>
> W3 & Q3.2. In Section 5.3, it says "Our scheduling algorithm identifies optimal assignments for all scenarios within 90 to 120 seconds". How is optimality defined here?
>
> Optimality is defined as the point when no further improvement in estimated throughput is observed after a certain number of iterations (approximately 20 in our case).

---

> ### Author Response · Authors · 2024-11-21
>
> W3 & Q3.3. Is the algorithm always guaranteed to find the optimal solution given enough search time?
>
> Our algorithm may not always be possible to find the absolute theoretical optimal solution. However, based on our scheduling algorithm, the optimization will iteratively narrow the gap between the current allocation and the theoretical optimal solution, where the iterative refinement process *addresses the limitations inherent in each phase*.
>
> The challenges in reaching optimal solutions lie in two aspects:
>
> - **In the graph partition phase,** creating an ideal graph partition in a single iteration is challenging since this phase lacks critical information (e.g., parallel strategy and KV cache communication path) from subsequent phases. Without these insights, the initial graph partitioning cannot guarantee an ideal utilization of the heterogeneous cluster, leading to potential communication bottlenecks and workload imbalances.
> - **The max flow phase** operates within the constraints set by the graph partition. The max-flow algorithm cannot achieve the theoretical maximum flow if the preceding graph partition results in suboptimal grouping. Limited inter-group communication bandwidth and unbalanced node capacities prevent the system from fully utilizing the network's data transfer capabilities.
>
> **Our iterative refinement approach.** *The iterative refinement phase is crucial in bridging the gap toward the optimal solutions.* It continuously evaluates and adjusts groupings, optimizes parallel configurations, and recalculates optimal KV cache communication paths based on updated partitions. This approach allows the algorithm to:
>
> - **Rebalance trade-offs for graph partition.** Balance intra-group resource optimization with inter-type communication efficiency for optimized resource utilization.
> - **Enhance max-flow potential.** Balance overutilized and underutilized edges within the formulated flow network for optimized data flow efficiency.
>
> Ultimately, this iterative approach incrementally moves the system closer to the optimal solutions of both resource utilization and data flow efficiency. While it may not always be possible to reach the absolute theoretical optimal solutions due to inherent system constraints, our method significantly narrows the gap.
>
> W3 & Q4. Also, how much overhead would the algorithm incur if running on a cluster with, for example, hundreds of GPUs? It may be hard to rent such a large number of GPUs for experiments, but evaluating algorithm overhead should be possible.
>
> Our algorithm incorporates elements like coarsening and projection operations specifically designed for handling large, complex heterogeneous clusters. Additionally, the max-flow guided edge swap helps overcome local minima and accelerates optimization, making the algorithm efficient for large graphs. We added more experiments on the scheduling algorithm about the algorithm running time and estimated throughput for different GPU cluster sizes. As demonstrated below.
>
> | Ngpus | Algorithm Convergence Time (min) |
> | --- | --- |
> | 64 | 4.03 |
> | 128 | 7.93 |
> | 192 | 21.66 |
> | 256 | 28.44 |
> | 320 | 47.77 |
>
> Experimental results show that our scheduling algorithm scales polynomially with cluster size and converges significantly faster than other heterogeneous scheduling algorithms, such as Helix [2], which takes around 50 minutes to search on 42 nodes. These findings highlight the potential of our algorithm to handle larger and more complex heterogeneous scheduling problems.
>
> We have integrated the analysis and experimental results into our updated paper (Appendix H).
>
> [2] Mei Y, Zhuang Y, Miao X, et al. Helix: Distributed Serving of Large Language Models via Max-Flow on Heterogeneous GPUs[J]. arXiv preprint arXiv:2406.01566, 2024.
>
> Q5. In the evaluation, it says HexGen-2 is compared against DistServe under the homogeneous setting with 8 H100 GPUs. Why is DistServe appearing in all heterogeneous settings in both Figure 6 and Figure 7?
>
> The central hypothesis of our experiments is:
>
> *What is the end-to-end performance comparison in terms of throughput and latency between
> HexGen-2 and the state-of-the-art homogeneous or heterogeneous generative inference systems?*
>
> This central hypothesis can be split into two aspects:
>
> 1. *What is the end-to-end performance comparison in terms of throughput and latency between HexGen-2 and HexGen under heterogeneous settings?*
> 2. *What is the end-to-end performance comparison in terms of throughput and latency between HexGen-2 under heterogeneous settings and DistServe under a homogeneous setting?*
>
> Our comparison with DistServe in Figures 6 and 7 addresses the second hypothesis.

---

> ### Author Response · Authors · 2024-11-27
>
> We sincerely appreciate the time and effort you have dedicated to reviewing our work!
>
> In our response and updated draft, we have provided more detailed explanations of the scheduling algorithm, included a small case study to illustrate the scheduling results, and added scalability experiments to better evaluate our algorithm.
>
> If there are any remaining concerns, we are fully committed to addressing them promptly and thoroughly. Thank you again for your patience and valuable insights. We look forward to further discussion.

---

### Official Review · Reviewer_iocs · 2024-11-02

**Soundness:** 2
**Presentation:** 2
**Contribution:** 3
**Rating:** 5
**Confidence:** 5

**Summary:**

This paper presents HexGen-2, a framework designed for serving language models on heterogeneous clusters with disaggregating prefill and decode computation on different devices. HexGen-2 first used a multi-round partition to divide all devices into multiple groups, and then introduced a maxflow algorithm to decide how to dispatch and route the workload on each device partition.
The authors did experiments on 5 different heterogeneous clusters, as well as 4 different workload pattern. Results show that even under a lower price budget, HexGen-2 matches the performance of the state-of-the-art disaggregated serving platform, DistServe, if not surpassing.

**Strengths:**

- The paper is the first to combine disaggregated serving together with heterogeneous devices, which is an important direction. This enhances the paper's novelty.
- The approach of using Max-Flow to solve the scheduling problem is interesting and efficient.

**Weaknesses:**

- The detail of the paper's method is not well motivated, neither well explained (to clarify physical node with abstract graph `nodes` in the algorithm, I use `this font` for abstract graph nodes):
  - In Graph Partition, the motivation of partitioning with two steps is not well explained. Specifically, it is confusing that why the first round minimizes the edge cost, while the second round maximizes the cost. Besides, why the `node` weight should be balanced (i.e., the computation capacity of each partition should be roughly the same)?
  - In Graph Partition, step 2, the concept of "partition" and "replica" causes confusion. Does each partitioning output group correspond to a replica? Is there multiple replicas for prefill, or only one replica for prefill, and one fore decide?
    - It seems to me that there are multiple prefill replicas (according to Max-Flow "for prefill model replicas, ..."). However, in this case, maximizing the total edge cost is not reasonable because the edges between prefill replicas are meaningless.
    - Does the algorithm consider the balance between node weights of (all) prefill replica(s) and that of (all) decode replica(s)? I think this is important because it avoids one part having too few devices to compute.
  - In Graph Partition, there is an input argument for the targeted group size $K$. How is this argument defined and set?
  - In Max-Flow, the author estimated the communication cost by dividing the total communication volume to the bandwidth. However, there exists a case that the bandwidth of some `node`s are shared, because these GPUs belong to the same physical node. On the other side, when both the send and receive side are on different physical nodes, the communication can be parallelized. Using the collective performance is not accurate.
  - In Max-Flow, the author mentioned that each "model replica" finds the optimal parallelism strategy themselves. Is this model replica inherited from Graph Partition? If so, what is the motivation of the projection?
  - In Max-Flow, what is the search space of the parallelism strategy? For example, when a model replica has two type of GPUs (e.g. one A100 and 4 V100), what is the parallelism strategy?
- The evaluation is not convincing to me. The baseline is too weak (see Question section for more details);

**Questions:**

Some questions that could help make the paper more clear is already mentioned in the Weakness section. In addition to that, there are some other concerns:
- According to Figure 4, the network topology is globally heterogeneous but locally homogeneous: to any destination device $u$, the bandwidth $(u_1,v),(u_2,v)$ from the same type of GPU $u_1$, $u_2$ is always the same. In this case, the Graph Partition algorithm seems an overkill to me: what if we consider to directly merge devices by the GPU types they belongs to, and then bipartition GPU types according to the inter-type bandwidth? Showing some nontrivial examples generated by HexGen-2 could also explain the importance of the graph partition algorithm.
- The context of serving with heterogeneous cluster lacks detailed introduction. As an important benchmark baseline, as well as the system that this work is built on top of, HexGen itself is not well introduced and explained. For example, the author mentioned the "genetic algorithm" in ablation study of scheduling algorithm but never explained it. Adding more detail to section 2 and 5 about the background of heterogeneous serving and the two specified baseline could help improve the paper's self-completeness.
- The benchmark baseline is too weak. On Llama-70B, vLLM, which does not even support disaggregation, reported a throughput of 25 req/s ($\approx$400 tok/s) on **4 H100-80GB** GPUs with the "prefill-heavy" workload (https://blog.vllm.ai/2024/09/05/perf-update.html). Given that their prefill token's average length is only around 400 tokens, I compare it with the "LPLD" benchmark: In this way, HexGen-2 uses 2x more budgets, but only with a maximum throughput around 550 (1.4x higher). Adding more experiments with state-of-the-art LLM serving platform for the homogeneous setup could better show the predominance of HexGen-2 against using homogeneous clusters with the same budget.

---

> ### Author Response · Authors · 2024-11-21
>
> W1. The detail of the paper's method is not well motivated, neither well explained (to clarify physical node with abstract graph `nodes` in the algorithm, I use `this font` for abstract graph nodes).
>
> Sorry for the confusion, in our scheduling algorithm, each graph node represents a physical GPU. And we enumerate the reply for each independent sub questions below:
>
> W1.1.1. Why the first round minimizes the edge cost, while the second round maximizes the cost?
>
> In our scheduling algorithm:
>
> - *The first round edge cost represents the inter group communication bandwidth.* We minimize it to maximize intra group communication bandwidth, thus optimizing each group’s processing capacity.
> - *The second round edge cost represents the inter partition KV cache communication bandwidth.* We maximize it to optimize KV cache communication.
>
> To clarify our scheduling algorithm, we present a simple example involving the partitioning of 8 heterogeneous GPUs.
>
> - **First round partition.** The 8 GPUs are partitioned into 4 groups, g1-4, each with 2 GPUs. We minimize the inter group edge cost to optimize each group’s processing capability.
> - **Second round partition.** The four groups are divided into two partitions, p1-2. g1-2 within p1 are prefill model replicas, g3-4 within p2 are decoding model replicas. We maximize the inter partition edge cost to *optimize KV cache communication efficiency*.
> - **Iterative refinement.** *Neither round guarantees global optimal partitioning.* The iterative refinement phase continuously adjusts the groups and partitions to achieve optimal results.
>
> W1.1.2. Why the node weight should be balanced (i.e., the computation capacity of each partition should be roughly the same).
>
> We want to gently point out that our approach focuses on balancing *memory capacity* rather than *computational capacity*. Here is the design consideration:
>
> **Why balance memory?** *The main objective is to avoid OOM issues and provide a good starting point for further optimization (iterative refinement).* Balancing compute capacity rather than memory often causes OOM issues in heterogeneous clusters, hindering the algorithm's convergence. Note that the node weight is not always balanced. *Node weights are balanced in the initial graph partition phase, but may become imbalanced to adapt to varying inference workloads in the iterative refinement phase.* For example, with light prefill and heavy decoding workloads, more node weights are assigned to the decoding model replicas to balance the resource needs across different inference phases.
>
> Again, thanks for pointing this out, we have integrated the clarification into our updated draft (Section 3.2).
>
> W1.2.1. In step 2 of graph partition, the concept of “partition” and “replica” causes confusion. It seems to me that there are multiple prefill replicas (according to Max-Flow "for prefill model replicas, ..."). However, in this case, maximizing the total edge cost is not reasonable because the edges between prefill replicas are meaningless.
>
> We summarize the difference between partition and replica below.
>
> - “partition” represents the collection of prefill or decoding model replicas;
> - “replica” represents a single prefill or decoding model replica.
>
> We want to gently point out that, in step 2, we focus on maximizing the edge cost *between prefill and decoding replicas*, rather than *between prefill replicas*. Concretely, the first partition divides the cluster into groups, each responsible for serving one model replica, the secondary partition further divides these groups into two partitions, p1 and p2, where groups within p1 are defined as prefill model replicas, and groups within p2 are defined as decoding model replicas. In this case, maximizing the edge cost between partitions (p1 and p2) is beneficial due to frequent KV cache transmissions between prefill and decoding model replicas.
>
> W1.2.2. Does the algorithm consider the balance between node weights of prefill and decoding replicas? I think this is important because it avoids one part having too few devices to compute.
>
> Yes, we considered that. Concretely:
>
> - **We balance node weights based on incoming workloads.** *Different workload types require varying node weight distributions between prefill and decoding model replicas.* For example, HPLD workloads with heavy prefill and light decoding demands require more node weights assigned to prefill model replicas to balance resource needs across different inference phases.
> - **We iteratively refine the node weight distribution.** *The iterative refinement phase described in Section 3.4 is responsible for the node weight adjustments.* Concretely, the max-flow guided edge swap operation identifies over- and under-utilized edges and swaps them to optimize node weight distribution, continuing until the optimal balance is achieved.

---

> ### Author Response · Authors · 2024-11-21
>
> W1.3. How to determine group size K?
>
> The initial group size, K, is determined by dividing the cluster's total memory capacity by the estimated memory needed for one model replica. If the group size is too large (too few GPUs per replica), some groups may lack sufficient memory, leading to OOM issues. Conversely, a small group size (too many GPUs per replica) increases communication overhead due to model parallelism. Note that *the group size K constantly changes during the iterative refinement phase***.** Initializing based on the memory requirement for one model replica provides an ideal starting point. The latter iterative refinement phase will optimize the results, group size and GPU allocation for each group are dynamically adjusted during the iterative refinement process. We have integrated the clarification into our updated draft (Section 3.2).
>
> W1.4. In Max-Flow, the author estimated the communication cost by dividing the total communication volume to the bandwidth. However, there exists a case that the bandwidth of some `node`s are shared, because these GPUs belong to the same physical node. On the other side, when both the send and receive side are on different physical nodes, the communication can be parallelized. Using the collective performance is not accurate.
>
> Thank you for pointing this out; it is a very interesting question.
>
> *We have found that the bandwidth-sharing problem you mentioned does not significantly affect our scheduling results.* To explain this, we first describe the scenario where bandwidth sharing occurs and then illustrate its effects on two types of cost estimations in our scheduling algorithm: KV cache communication cost estimation and parallel strategy communication cost estimation.
>
> - **Bandwidth sharing.** *Bandwidth sharing only occurs in inter-machine communication.* Concretely, bandwidth sharing primarily affects inter-machine low-bandwidth links like TCP/Ethernet connections. Within a single machine, GPUs communicate over high-bandwidth links such as PCIe and NVLink, where bandwidth sharing is negligible.
> - **KV cache communication cost estimation.** *KV cache communication is on high bandwidth only*. Our scheduling algorithm always routes KV cache communications through high-bandwidth links like NVLink and PCIe to prevent system bottlenecks, as efficient KV cache communication is essential in disaggregated inference architectures requiring high bandwidth [1,2]. Thus, bandwidth-sharing issues on low-bandwidth inter-machine links do not affect the estimation of our KV cache communication costs.
> - **Parallel strategy cost estimation.** *While bandwidth sharing in inter-node communications may introduce some inaccuracies in cost estimation, it has minimal impact on scheduling decisions.* Given the significant bandwidth disparity between intra-node (PCIe/NVLink) and inter-node (Ethernet/TCP) links, our scheduling algorithm consistently routes minimal communication volumes (e.g., inter-stage pipeline communication) through low-bandwidth links to maintain efficient parallelism, regardless of bandwidth sharing considerations.
>
> We acknowledge that considering the bandwidth-sharing problem could enhance the accuracy of our communication cost estimations.
>
> [1] Zhong Y, Liu S, Chen J, et al. {DistServe}: Disaggregating Prefill and Decoding for Goodput-optimized Large Language Model Serving[C]//18th USENIX Symposium on Operating Systems Design and Implementation (OSDI 24). 2024: 193-210.
>
> [2] Patel P, Choukse E, Zhang C, et al. Splitwise: Efficient generative llm inference using phase splitting[C]//2024 ACM/IEEE 51st Annual International Symposium on Computer Architecture (ISCA). IEEE, 2024: 118-132.
>
> W1.5. In Max-Flow, the author mentioned that each "model replica" finds the optimal parallelism strategy themselves. Is this model replica inherited from Graph Partition? If so, what is the motivation of the projection?
>
> Yes, the model replica in max-flow is inherited from the graph partition.
>
> *The purpose of the projection step is to recover the GPU information (e.g., GPU count, type, and communication bandwidth) from a super node*, which is then used to determine the parallel strategy. Concretely, during coarsening, all GPUs within a model replica are merged into a super node, which conceals information about GPU count, type, and communication bandwidth within the model replica. The projection operation reverses this coarsening process to recover the GPU details from the super node, which are then used to determine the optimal strategy.

---

> ### Author Response · Authors · 2024-11-21
>
> W1.6. In Max-Flow, what is the search space of the parallelism strategy? For example, when a model replica has two types of GPUs (e.g. one A100 and 4 V100), what is the parallelism strategy.
>
> In our formulation, the search space for parallelism strategies includes all the viable combinations of tensor model parallelism (TP) and pipeline parallelism (PP) that can be configured given the constraints of the total number of GPUs.
>
> Consider the example with one A100 GPU and four V100 GPUs, the possible configurations are (TP=1, PP=5) and (TP=5, PP=1). For a setup with four A100 GPUs and four V100 GPUs, the possible configurations include (TP=1, PP=8), (TP=2, PP=4), (TP=4, PP=2), and (TP=8, PP=1).
>
> Q1. The Graph Partition algorithm seems an overkill to me: what if we consider to directly merge devices by the GPU types they belong to, and then bipartition GPU types according to the inter-type bandwidth? Showing some nontrivial examples generated by HexGen-2 could also explain the importance of the graph partition algorithm.
>
> Thanks for providing this insightful consideration, we want to clarify that directly merging devices by GPU type can lead to two major issues:
>
> - **Imbalance**: Different GPU types have varying memory limits and compute power, causing imbalances in the cluster. For example, with 2 A100, 2 A6000, and 2 L40 GPUs serving a Llama-70B model, grouping by type would cause OOM issues for A6000 and L40 GPUs. In contrast, a memory-balanced graph partition would assign 1 A100, 1 A6000, and 1 L40 to serve each prefill and decoding model replica.
> - **Restricted optimization**: Merging GPUs by type limits the algorithm's ability to find an optimal plan. For instance, with 4 A100 and 4 A6000 GPUs connected via Ethernet to serve a Llama-70B model, HexGen-2 finds that assigning 2 A100 and 2 A6000 per replica balances prefill and decoding while ensuring efficient KV cache communication via NVLink and PCIe. In contrast, merging by type (4 A100s and 4 A6000s separately) can lead to poor system performance due to significant differences in the compute and memory capacities of the prefill and decoding model replicas, as well as inefficient KV cache communication over Ethernet.
>
> **Scalability.** *We chose the graph partitioning algorithm primarily for its scalability*. In larger, complex heterogeneous environments, simple heuristics like grouping by GPU type become impractical. Graph partition algorithm scales efficiently. Iterative refinement with coarsening and projection minimizes re-partition overhead, ensuring the algorithm performs efficiently as GPU count and interconnections grow.
>
> Additionally, we also conducted experiments on larger clusters to evaluate the scalability of our scheduling algorithm. The results indicate that the algorithm scales polynomially and shows potential for addressing more complex heterogeneous scheduling challenges. We have integrated this case study into our updated draft (Appendix H).
>
> | **Ngpus** | **Algorithm Convergence Time (min)** |
> | --- | --- |
> | 64 | 4.03 |
> | 128 | 7.93 |
> | 192 | 21.66 |
> | 256 | 28.44 |
> | 320 | 47.77 |
>
> Q2. The context of serving with heterogeneous cluster lacks detailed introduction. As an important benchmark baseline, as well as the system that this work is built on top of, HexGen itself is not well introduced and explained. For example, the author mentioned the "genetic algorithm" in ablation study of scheduling algorithm but never explained it. Adding more detail to section 2 and 5 about the background of heterogeneous serving and the two specified baseline could help improve the paper's self-completeness.
>
> Thanks for your suggestions. We have added more descriptions of the genetic algorithm and two specified baselines in the updated draft (Section 5.1 and 5.3), as listed below:
>
> - “we compare HexGen-2 with DistServe as the state-of-the-art approach under the homogeneous setting, which enhances LLM serving by disaggregating prefill and decoding computations across different GPUs, allowing different resource allocation and parallelism for each phase. And HexGen as the state-of-the-art approach under heterogeneous settings, which is a distributed inference engine that efficiently manages LLM inference across heterogeneous environments, leveraging asymmetric parallelism with a scheduling algorithm to optimize resource allocation.”
> - “The genetic algorithm, designed to optimize model deployment, uses a population-based approach involving merge, split, and swap operations to iteratively refine GPU groupings. In our comparison, we replaced the group generation step in the graph partition phase and the iterative refinement phases of our algorithm with the genetic algorithm to enable HEXGEN-2 with this method.”

---

> ### Author Response · Authors · 2024-11-21
>
> W2 & Q3. The benchmark baseline is too weak. On Llama-70B, vLLM, which does not even support disaggregation, reported a throughput of 25 req/s (400 tok/s) on **4 H100-80GB** GPUs with the "prefill-heavy" workload (https://blog.vllm.ai/2024/09/05/perf-update.html). Given that their prefill token's average length is only around 400 tokens, I compare it with the "LPLD" benchmark: In this way, HexGen-2 uses 2x more budgets, but only with a maximum throughput around 550 (1.4x higher). Adding more experiments with state-of-the-art LLM serving platform for the homogeneous setup could better show the predominance of HexGen-2 against using homogeneous clusters with the same budget.
>
> To demonstrate the superiority of HexGen-2 over other state-of-the-art LLM serving platforms, we included vLLM as a baseline. The vLLM benchmarks described in https://blog.vllm.ai/2024/09/05/perf-update.html assume all requests arrive simultaneously—specifically, 500 requests at once—allowing for maximum batching and optimal throughput in a controlled environment. In contrast, real-world traces involve requests arriving sequentially at a certain rate. In our experiments, we scaled the arrival rates of the Azure Conversation dataset to 8-16 requests per second, following the same experimental setup as [1]. We evaluate vLLM under the same conditions as our experiments to ensure a fair comparison. The experimental results are summarized below:
>
> |  |  | **HPLD** | **HPHD** | **LPHD** | **LPLD** | **Online** |
> | --- | --- | --- | --- | --- | --- | --- |
> | **Heterogeneous Setting 1** | HexGen-2 | 157 tokens/s | 448 tokens/s | 689 tokens/s | 570 tokens/s | 350 tokens/s |
> | **Heterogeneous Setting 1** | HexGen | 123 tokens/s | 375 tokens/s | 492 tokens/s | 407 tokens/s | 259 tokens/s |
> | **Homogeneous Setting** | DistServe | 128 tokens/s | 368 tokens/s | 553 tokens/s | 291 tokens/s | 251 tokens/s |
> | **Homogeneous Setting** | vLLM | 97 tokens/s | 437 tokens/s | 563 tokens/s | 270 tokens/s | 256 tokens/s |
>
> We have integrated the experimental results into our updated draft (Section 5.2 and Appendix F).
>
> [1] Zhong Y, Liu S, Chen J, et al. {DistServe}: Disaggregating Prefill and Decoding for Goodput-optimized Large Language Model Serving[C]//18th USENIX Symposium on Operating Systems Design and Implementation (OSDI 24). 2024: 193-210.

---

> ### Author Response · Authors · 2024-11-27
>
> We sincerely appreciate the time and effort you have dedicated to reviewing our work!
>
> In our response and updated draft, we have added detailed explanations of our scheduling algorithm, the background of heterogeneous serving, and additional baseline experiments (vLLM).
>
> If there are any unresolved concerns, we are fully committed to addressing them promptly and to the best of our ability. Thank you again for your patience and valuable insights. We look forward to further discussion.

---

> ### Comment · Reviewer_iocs · 2024-11-27
> **Feedback to rebuttal**
>
> Dear authors,
>
> Thanks for your response. Below is my feedback
>
> ---
>
> W 1.1
>   - 1.1.2 I acknowledge the importance of avoiding OOM. However, the prefill stage is still compute intense, making compute capacity still holds its importance.
>
> W 1.2
>   - 1.2.2 It seems like such a workload balance is not highlighted in the main text. How is the node weight distribution determined? Is it by some profiling of requests?
>
> W 1.3
>   - How is the new K computed during the local refinement?
>   - When estimating the number of replicas based on the memory required for each replica, I'd assume KV-cache an important factor for memory estimation. How is this factor estimated?
>
> W 1.4 The optimization target of the first round partition is to minimize the inter-group communication (i.e. maximizing the intra-group communication). In this way, the high-bandwidth is more likely used for intra-group communication, which is supposed to be the model parallel. The rebuttal seems to suggest that neither KV-transmission nor model parallel communication would use the low-bandwidth communication, which is confusing to me.
>
> W 1.5 The rebuttal addressed this concern.
>
> W 1.6 Given the fact that a group may have multiple type of GPUs, simply searching for different degrees might not be enough. Using A100 and V100 for the same number of layers could be a waste of the better GPUs.
>
> Q2 The rebuttal addressed this concern.
>
> W2, Q3 The rebuttal addressed this concern.
>
> ---
>
> I also noticed that the citation format is incorrect. There are many missing brackets.

---

> > ### Author Response · Authors · 2024-12-01
> >
> > Dear reviewer,
> >
> > Thank you again for your previous insightful feedback! As the discussion time is coming to an end, we would greatly appreciate it if you could check our newest response. If there are further concerns, we will try our best to address them. Thank you very much for your time!

---

> ### Author Response · Authors · 2024-11-28
>
> Thank you very much for your detailed response regarding your concerns! We will address each of your questions below. If you have any further concerns, please don't hesitate to let us know.
>
> W 1.1
> - 1.1.2 I acknowledge the importance of avoiding OOM. However, the prefill stage is still compute intense, making compute capacity still holds its importance.
>
> >This understanding is absolutely correct—compute capacity is very crucial in this estimation; actually, we did not only consider the memory-balanced case throughout this optimization procedure — Note that the algorithm begins from a promising starting point (memory-balanced partition) to avoid OOM issues and iterative refines towards an optimal point, typically between memory-balanced and compute-balanced states. **When determining the initial status, only considering memory constraint help us avoid some suboptimal initial points in the search space, but throughout the optimisation procedure, computational factor contributes significantly in determining the final search result**.
> >
> >For example:
> >- In the first phase, we have a prefill and a decoding model replica that are memory-balanced but not compute-balanced.
> >- In the second phase, the max-flow algorithm attempts to maximize network flow. Due to different compute capacities, it finds that the prefill model replica is over-utilized while the decoding model replica is under-utilized.
> >- In the iterative refinement phase, the scheduling algorithm will swap GPUs from the under-utilized decoding model replica to the over-utilized prefill model replica to improve the overall flow.
> >
> >This iterative refinement approach starts from a promising state to avoid OOM issues and gradually moves towards an optimal balance.
>
> W 1.2
> - 1.2.2 It seems like such a workload balance is not highlighted in the main text. How is the node weight distribution determined? Is it by some profiling of requests?
>
> >Thanks for the further clarification of the question. We were mentioning the discussion of such workload balance in section 3.4 (iterative refinement phase):
> >
> >”This swap operation is essential in terms of: (ii) adjusting the node and edge weights across intra-groups to optimize resource allocation.”
> >
> >We do **NOT** need to determine the node weight distribution manually (thus, we do not introduce any profiling before the scheduling). Concretely, the node weight distribution is automatically adjusted based on the flow assignment from the max-flow phase. This flow assignment indicates which node is underutilized and which node is overutilized in the flow network, and the iterative refinement phase will adjust the node weight distribution based on this information.
> >
> >For example, assume we have one prefill and one decoding model replica:
> >- The flow assignment in the second phase indicates that the prefill model replica’s used/total capacity is **100/100**, while the decoding model replica’s used/total capacity is **100/150**.
> >- The scheduling algorithm indicates that the prefill model replica is overutilized (the capacity is fully used), while the decoding model replica is underutilized (only 2/3 of the total capacity is used).
> >- The scheduling algorithm will try to allocate more GPUs to the prefill model replica to see if a higher max flow can be achieved based on the new resource allocation.
> >
> >This process indirectly influences the node weight distribution (more weights are assigned to the prefill model replica).
>
> W 1.3
> - How is the new K computed during the local refinement?
>
> >The K does not need to be determined explicitly during the iterative refine phase; it is automatically tuned based on the refinement decision.
> >
> >Assume we have two groups, g1 and g2. If local refinement decides to swap some GPUs out of g1, the decision can either be:
> >
> >- Move these GPUs to g2.
> >- Use these GPUs to form a new group g3.
> >
> >After this refinement, the scheduling will go through the first and second phases again (i.e., determine the prefill and decoding model serving groups, parallel strategy, and KV cache communication strategy) and see if a higher max flow could be achieved based on the new partition.
> >
> >In the first case, the number of groups remains 2; in the second phase, it increases to 3. Thus, K is determined by the local refinement process rather than being a fixed value.

---

> ### Author Response · Authors · 2024-11-28
>
> W 1.3
> - When estimating the number of replicas based on the memory required for each replica, I'd assume KV-cache an important factor for memory estimation. How is this factor estimated?
>
> >In our memory estimation function is illustrated in Appendix A, Table 1, where the term $2b_{t}(s_{t}^{\text{in}}+s_{t}^{\text{out}})HB_{\text{type}}$ estimates the KV cache memory ($b_t$ is the batch size). To estimate the total KV cache memory, we typically set $b_t$ to 32, assuming a batch size of 32 concurrent requests. Thus the estimated memory for single model replica should be: model parameter size + 32 * single request KV cache size. We just include this in the updated draft.
>
> W 1.4 The optimization target of the first round partition is to minimize the inter-group communication (i.e. maximizing the intra-group communication). In this way, the high-bandwidth is more likely used for intra-group communication, which is supposed to be the model parallel. The rebuttal seems to suggest that neither KV-transmission nor model parallel communication would use the low-bandwidth communication, which is confusing to me.
>
> >Sorry for the confusion. We want to clarify further that:
> >
> >KV transmission will not use low-bandwidth links. Even if the initial partition allocates high-bandwidth links within a group, the edge swap operation in the iterative refinement phase ensures that high-bandwidth links are reassigned for KV transmission. As mentioned in section 3.4:
> >
> >“This swap operation is essential in terms of: i) balancing the inter- and intra-group
> edge weights to maintain high intra-group capacities while enabling efficient inter-group KV cache communicating;”
> >
> >The low-bandwidth links will either **be avoided** or used for **pipeline parallelism**. Note that pipeline parallelism only communicates layer activations between stages via send-receive operations, resulting in lower communication volume. Thus, to ensure efficient parallel inference, if there are low bandwidth links within a model serving group, the scheduling algorithm always uses pipeline parallelism for low-bandwidth links to minimize communication overhead.
>
> W 1.6 Given the fact that a group may have multiple type of GPUs, simply searching for different degrees might not be enough. Using A100 and V100 for the same number of layers could be a waste of the better GPUs.
>
> >We apologize for not mentioning the imbalanced layer partitioning in pipeline parallelism in addition to allocating different degrees in our approach in last reply. We illustrate how we leverage the layer partitioning to address the issue by the an example with A100 and V100  GPUs. If we have one A100 (80 GB) and four V100s (each 48 GB), approximately 30% of the layers are allocated to the A100, while the remaining 70% are distributed evenly among the four V100s.
>
> I also noticed that the citation format is incorrect. There are many missing brackets.
>
> > Sorry for this incorrect format; we will update the correct version of the draft.

---

### Official Review · Reviewer_cxpP · 2024-11-03

**Soundness:** 3
**Presentation:** 4
**Contribution:** 3
**Rating:** 8
**Confidence:** 3

**Summary:**

The paper introduces HexGen2, a distributed LLM inference framework targeting heterogeneous GPU clusters. The framework disaggregates prefilling and decoding tasks onto different GPUs, ensuring that the two phases do not interfere with each other and that they can be parallelized in different ways when doing so is beneficial.

**Strengths:**

- LLM inference on heterogeneous GPUs is a critical and timely issue. Separating the prefill and decode stages when serving LLMs in diverse clusters represents a novel approach.
- Evaluation is thorough
- Very clear presentation

**Weaknesses:**

- The evaluation does not include an ablation study that compares HexGen2's runtime with those of DistServe and HexGen. Understanding HexGen2's performance in terms of throughput and latency compared to DistServe and HexGen in a homogeneous setting would be valuable.
- In the introduction, the authors assert that disaggregated inference is the most efficient framework for serving large language models (LLMs) without providing proof or citations. Is this assertion already widely accepted by the community, or is there still an ongoing debate regarding the advantages of disaggregation versus chunked prefills? It would be beneficial for the authors to clarify this—at least in the related work section—and to explain their reasoning for choosing the disaggregation approach over chunked prefills.

**Questions:**

- Is HexGen2 built on an existing runtime, or was it developed from scratch? Specifically, how does HexGen2's runtime compare to those of DistServe and HexGen? Understanding this would be helpful for better assessing the end-to-end performance comparison with the other two frameworks.
- In the evaluation settings, could you explain your motivation for selecting a 70% lower budget as the target for the evaluation scenario? How did you arrive at this percentage?
- Do you have any insights or, preferably, evaluation data regarding which additional heterogeneous clusters HexGen2 would perform well with, besides those already evaluated? Does HexGen2 support any type of cluster, or are there specific restrictions regarding GPU types, interconnects, or CUDA architectures?

---

> ### Author Response · Authors · 2024-11-21
>
> W1 & Q1. Compare HexGen2, DistServe, and HexGen in a homogeneous setup.
>
> We conduct a set of additional experiments in a homogeneous setup. We enumerate the setup and results below:
>
> To compare the runtime of HexGen2 with DistServe and HexGen, we rented 4 H100 GPUs from the RunPod platform and tested serving throughput on the OPT-30B model using the four types of LLM inference workloads described in Section 5.1.
>
> **Compare with DistServe.** We found that for certain inference workloads, the scheduling results of HexGen2 and DistServe differ. For example, with the HPLD workload, HexGen2 favors replicating more model replicas to enhance the system's parallel processing, while DistServe prefers model parallelism to distribute the computation of a single model replica across multiple GPUs. Experimental results demonstrate that HexGen2 outperforms DistServe in certain cases due to better scheduling results while delivering comparable performance when the scheduling outcomes are the same.
>
> **Compare with HexGen.** HexGen2, with optimized scheduling in a disaggregated architecture, minimizes interference between the prefill and decoding phases of LLM inference. It selects appropriate parallelism and batching strategies for each phase, resulting in improved inference performance compared to HexGen in a homogeneous environment.
>
> |  | **HexGen2** | **DistServe** | **HexGen** |
> | --- | --- | --- | --- |
> | **HPLD** | 365 tokens/s | 302 tokens/s | 277 tokens/s |
> | **HPHD** | 683 tokens/s | 692 tokens/s | 505 tokens/s |
> | **LPHD** | 758 tokens/s | 774 tokens/s | 533 tokens/s |
> | **LPLD** | 730 tokens/s | 553 tokens/s | 545 tokens/s |
>
> We have incorporated the ablation study into our updated draft (Appendix G).
>
> W2. Comparison of the advantages of disaggregation versus chunked prefills.
>
>  Different from disaggregated inference paradigm, chunked prefill is a method that divides input tokens into smaller chunks, which are then processed in a continuous batch. Chunked prefill approach simplifies scheduling by treating all nodes uniformly and enhances computational efficiency during decoding, potentially improving machine utilization. However, chunked prefill may not result in significant performance gains across all workload types. We conduct a small set of additional experiments to evaluate chunked prefill using vLLM on one H100 GPU serving the OPT-30B model. Experimental results demonstrate that on HPLD and LPLD workloads, chunked prefill brings an approximately 20% throughput improvement, while it only brings around 5% throughput gains on HPHD and LPHD workloads. Therefore, we choose disaggregation, which enables different batching strategies, resource allocations, and parallel approaches for each phase, providing greater flexibility in handling various types of workloads.
>
> We have incorporated all the detailed discussion into our updated draft (Appendix D).

---

> ### Author Response · Authors · 2024-11-21
>
> Q1. Is HexGen2 built on existing runtime or was it developed from scratch?
>
> To be concrete, we developed HexGen2 on top of HexGen, and our scheduling results are fully compatible with other disaggregating frameworks such as DistServe.
>
> Q2. In the evaluation settings, could you explain your motivation for selecting a 70% lower budget as the target for the evaluation scenario? How did you arrive at this percentage?
>
> In the full-budget heterogeneous scenario, our framework largely outperforms the homogeneous case. This raises an interesting question for us: *What is the minimum budget required for a heterogeneous setup to match the performance of a full-budget homogeneous setup?* Thus, we tested on different budgets, and the results demonstrate that we can reduce the budget by up to 30% in the heterogeneous case while achieving comparable performance to the full-budget homogeneous case. Thus, 70% budget is ideal for demonstrating the cost efficiency of serving on heterogeneous clusters.
>
> Q3.1. Do you have any insights or, preferably, evaluation data regarding which additional heterogeneous clusters HexGen2 would perform well with, besides those already evaluated?
>
> We tend to believe *integrating more cost-effective GPUs into the heterogeneous cluster could potentially optimize serving performance.* For example, for workloads with high bandwidth demands, such as heavy decoding jobs, A100 GPUs are more cost-efficient than H100, as the H100's HBM bandwidth is 1.64 times greater than the A100's, but at double the price. Therefore, a heterogeneous cluster with more A100 GPUs and fewer H100 GPUs could potentially provide better serving performance for decoding-intensive workloads.
>
> Q3.2. Does HexGen2 support any type of cluster, or are there specific restrictions regarding GPU types, interconnects, or CUDA architectures?
>
> Currently, HexGen2 supports only NVIDIA GPUs but is compatible with any type, without requiring specific interconnects or CUDA architectures. One potential interesting direction is to explore the deployment of our framework over AI-chips from different vendors (e.g., AMD GPU, TPU, NPU, etc.). The current main limitation is the lack collective communication support over heterogenous AI-chips from different vendors, we see some very recent works from the network system community attempting to solve this problem, we leave this as an interesting future work for HexGen2.

---

> ### Author Response · Authors · 2024-11-27
>
> We sincerely appreciate the time and effort you have dedicated to reviewing our work!
>
> In our response and updated draft, we have added more experiments comparing homogeneous setups, included discussions on disaggregation versus chunked prefill, and clarified various details.
>
> If there are any remaining concerns, we are fully committed to addressing them promptly and thoroughly. Thank you again for your patience and valuable insights. We look forward to further discussion.

---

### Official Review · Reviewer_cEk2 · 2024-11-04

**Soundness:** 3
**Presentation:** 3
**Contribution:** 2
**Rating:** 6
**Confidence:** 3

**Summary:**

The paper introduces the disaggregated LLM servings into the heterogeneous environment. The deployment is formulated as an optimization problem, a graph max flow problem. Results show improvement compared to the homogeneous methods.

**Strengths:**

- The idea of scheduling among heterogeneous GPUs is intuitive. And the design provides insights. Using graph partitioning and modeling the query processing as max-flow problem is valid.

- The paper is well-written and generally easy to follow.

**Weaknesses:**

- Perhaps the paper can provide some analysis about the optimization problem.
  - Is there an upper bound of the performance (from the perspective of graph partitioning and max flow problem, respectively) your approach can reach? How is the performance your method achieves compared to this upper bound?

- The main contribution seems to be formulating the optimization problem while some parts of the solutions are not that new.

- Some details can be clarified in the paper.
  - Which GPU is used in Figure 1?
  - What is the outcome (objective) of the graph partitioning? How to determine which group is prefill group and which one is decoding group? Are the results optimal?

- Miscellaneous
  - carefully use \cite and \citep for citation.
  - Capitalize the first letter after the colon.

**Questions:**

See above.

---

> ### Author Response · Authors · 2024-11-21
>
> W1.1. Perhaps the paper can provide some analysis about the optimization problem.
>
> We appreciate the suggestions and make the corresponding changes in the updated draft (Appendix C). The main updates are summarized below:
>
> **Optimization problem overview.** The scheduling algorithm aims to optimize the deployment of large language model (LLM) inference workloads on a heterogeneous GPU cluster. The optimization involves the following essential phases.
>
> - **Graph partition.** The initial partition focuses on creating memory-balanced groups and optimizing the capacity within each group. The secondary partition determines group type (i.e., prefill or decoding), focusing on maximizing inter-type communication bandwidth for efficient KV cache transfer.
> - **Max-flow.** This phase determines optimal parallel strategies for each group and determines the optimal inter-type KV cache communication paths based on the max-flow outputs.
> - **Iterative Refinement.** This phase continuously adjusts partitions and strategies based on workload demands until no further improvements can be made.
>
> W1.2. Is there an upper bound of the performance (from the perspective of graph partitioning and max flow problem, respectively) your approach can reach?
>
> Yes, there are upper bounds of the performance for both graph partition and max-flow phases.
>
> **The upper bound for graph partitioning** indicates the *optimal utilization of heterogeneous computation power and connections.* The theoretical upper bound of the graph partition phase is achieved when the cluster is partitioned into groups with balanced memory capacities and optimized processing capabilities, and the groups are assigned types (i.e., prefill or decoding) in a manner that maximizes inter-type communication bandwidth for key-value (KV) cache transfers.
>
> **The upper bound for max-flow** indicates the *maximum possible data flow within the cluster*. The theoretical upper bound of the max flow phase is determined by the maximum possible data transfer rate of the entire system. This upper limit is achieved when the system fully utilizes the inter-type network bandwidth for KV cache transfers and optimizes the processing capabilities of the prefill and decoding model replicas.
>
> W1.3. How is the performance your method achieves compared to this upper bound?
>
> Based on our scheduling algorithm, the optimization will iteratively narrow the gap between the current allocation and the theoretical upper bounds, where the iterative refinement process *addresses the limitations inherent in each phase*.
>
> The challenges in reaching upper bounds lie in two aspects:
>
> - **In the graph partition phase,** creating an ideal graph partition in a single iteration is challenging since this phase lacks critical information (e.g., parallel strategy and KV cache communication path) from subsequent phases. Without these insights, the initial graph partitioning cannot guarantee an ideal utilization of the heterogeneous cluster, leading to potential communication bottlenecks and workload imbalances.
> - **The max flow phase** operates within the constraints set by the graph partition. The max-flow algorithm cannot achieve the theoretical maximum flow if the preceding graph partition results in suboptimal grouping. Limited inter-group communication bandwidth and unbalanced node capacities prevent the system from fully utilizing the network's data transfer capabilities.
>
> **Our iterative refinement approach.** *The iterative refinement phase is crucial in bridging the gap toward the upper bounds.* It continuously evaluates and adjusts groupings, optimizes parallel configurations, and recalculates optimal KV cache communication paths based on updated partitions. This approach allows the algorithm to:
>
> - **Rebalance trade-offs for graph partition.** Balance intra-group resource optimization with inter-type communication efficiency for optimized resource utilization.
> - **Enhance max-flow potential.** Balance overutilized and underutilized edges within the formulated flow network for optimized data flow efficiency.
>
> Ultimately, this iterative approach incrementally moves the system closer to the upper limits of both resource utilization and data flow efficiency. While it may not always be possible to reach the absolute theoretical upper bounds due to inherent system constraints, our method significantly narrows the gap.

---

> ### Author Response · Authors · 2024-11-21
>
> W2. The main contribution seems to be formulating the optimization problem while some parts of the solutions are not that new.
>
> Thanks for this feedback. We agree that graph partitioning and max-flow algorithms are classic optimization methods. On the other hand, we want to gently suggest that our main contribution lies in the novel integration and adaptation of these techniques to address the unique challenges of scheduling disaggregated LLM serving on heterogeneous clusters.
>
> W3.1 Details can be clarified in the paper. "Which GPU is used in Figure 1？"
>
> We are sorry for the confusion and clarify it here. The GPU we used in Figure 1 is an A100 SXM GPU with 80 GB VRAM and 16 vCPU rent from the RunPod platform. We’ve integrated this detail into our updated paper.
>
> W3.2 What is the outcome (objective) of the graph partitioning? How to determine which group is prefill group and which one is decoding group? Are the results optimal?
>
> **Objectives of graph partitioning.** Our graph partitioning has two objectives. The first objective is to find the GPU group construction (each group is responsible for serving one model replica), and the second objective is to find the type of each group (which group is responsible for prefill and which group is responsible for decoding).
>
> **How to determine the prefill and decoding group.** Step 1 (initial partition) in graph partitioning divides the cluster into multiple model serving groups by minimizing inter-group communication bandwidth and balancing memory capacity. Step 2 (secondary partition) further divides these model serving groups into two partitions: all model replicas within the first partition are determined as prefill model replicas, and all model replicas within the second partition are determined as decoding model replicas. The inter-partition communication bandwidth is used to transmit the KV cache between prefill and decoding model replicas, which is why the secondary partition aims to maximize inter-partition communication bandwidth for efficient KV cache transmission.
>
> **Are the results optimal?** *The initial graph partitioning outcomes (i.e., group partition and type) may not be optimal, but they will be refined to be optimal.* Concretely, the group partition and type determined by the graph partitioning algorithm will be continuously refined during the iterative refinement phase to ensure optimal final results.
>
> W4. Miscellaneous.
>
> Thanks for pointing this out. We have updated the draft to address the issues you mentioned.

---

> ### Author Response · Authors · 2024-11-27
>
> We sincerely appreciate the time and effort you have dedicated to reviewing our work!
>
> In our response and updated draft, we have provided a more detailed analysis of the optimization problem and clarified key details of the paper.
>
> If there are any remaining concerns, we are fully committed to addressing them promptly and thoroughly. Thank you again for your patience and valuable insights. We look forward to further discussion.

---

### Author Response · Authors · 2024-11-21

# **Summary**

We thank all the reviewers for their valuable comments. All reviews acknowledge the novelty of our paper in combining disaggregated serving together with heterogeneous devices and recognize that it represents an important direction in LLM serving. The noted strengths include insightful system design, interesting and efficient scheduling algorithm, thorough evaluation, and good improvements over prior works.

**Current concerns:** the concerns about the current draft mainly involve two aspects:

- Explanation of motivation and details of the scheduling algorithm.
- Evaluation and comparison of HexGen-2 with more baselines.

In order to resolve these two issues, we have made the following efforts:

- We have added additional discussion about the motivation and details of our scheduling algorithm to our updated draft.
- We have provided additional experimental results to address concerns about our evaluations. This includes incorporating more baselines (e.g., vLLM), experimenting with HexGen-2 in the homogeneous setting, and analyzing the scalability of our scheduling algorithm.

We have updated draft and appreciate it if reviewers would gently check the updated version of our paper.

---

### Author Response · Authors · 2024-12-02

Dear Reviewers,

Today marks the final day of our discussion period. If you have any further concerns or suggestions, please don't hesitate to share them. Your feedback is greatly appreciated and will help enhance our work.

Thank you for your invaluable advice and support.

---

### Meta-Review · Area_Chair_A7gi · 2024-12-23

**Metareview:**

This paper presents HexGen-2, a framework designed for serving language models on heterogeneous clusters with disaggregating prefill and decode computation on different devices. It seems to be the first work to combine disaggregated prefill/decode and heterogeneous GPUs. The author did experiments on different cluster setups and showed advantages over baselines including the latest DistServe.

**Additional Comments On Reviewer Discussion:**

Reviewers raised concerns about the clarity of the presentation, and details about the evaluation; The authors have provided a good rebuttal in addressing most of the concerns. Overall I find the paper a nice addition to the line of research on disaggregated serving of LLMs.

---

### Decision · Program_Chairs · 2025-01-22

Accept (Poster)